# Efficient Energy Management of IoT-Enabled Smart Homes Under Price-Based Demand Response Program in Smart Grid

**DOI:** 10.3390/s20113155

**Published:** 2020-06-02

**Authors:** Ghulam Hafeez, Zahid Wadud, Imran Ullah Khan, Imran Khan, Zeeshan Shafiq, Muhammad Usman, Mohammad Usman Ali Khan

**Affiliations:** 1Department of Electrical and Computer Engineering, COMSATS University Islamabad, Islamabad 44000, Pakistan; ghulamhafeez393@gmail.com or; 2Department of Electrical Engineering, University of Engineering and Technology, Mardan 23200, Pakistan; imran@uetmardan.edu.pk (I.K.); zeeshan@uetmardan.edu.pk (Z.S.); 3Department of Computer Systems Engineering, University of Engineering and Technology Peshawar, Peshawar 25000, Pakistan; zahidmufti@nwfpuet.edu.pk; 4College of Underwater Acoustics Engineering Harbin Engineering University Heilongjiang, Harbin 150001, China; 5Department of Computer Software Engineering, University of Engineering and Technology, Mardan 23200, Pakistan; usman@uetmardan.edu.pk; 6Department of Electrical Engineering, University of Engineering and Technology, Peshawar 25000, Pakistan; usman.ali@uetpeshawar.edu.pk

**Keywords:** energy management, internet-of-things, residential building, sensors, smart appliances, price-based demand response programs, scheduling, smart grid

## Abstract

There will be a dearth of electrical energy in the prospective world due to exponential increase in electrical energy demand of rapidly growing world population. With the development of internet-of-things (IoT), more smart devices will be integrated into residential buildings in smart cities that actively participate in electricity market via demand response (DR) programs to efficiently manage energy in order to meet this increasing energy demand. Thus, with this incitement, an energy management strategy using price-based DR program is developed for IoT-enabled residential buildings. We propose a wind-driven bacterial foraging algorithm (WBFA), which is a hybrid of wind-driven optimization (WDO) and bacterial foraging optimization (BFO) algorithms. Subsequently, we devised a strategy based on our proposed WBFA to systematically manage the power usage of IoT-enabled residential building smart appliances by scheduling to alleviate peak-to-average ratio (PAR), minimize cost of electricity, and maximize user comfort (UC). This increases effective energy utilization, which in turn increases the sustainability of IoT-enabled residential buildings in smart cities. The WBFA-based strategy automatically responds to price-based DR programs to combat the major problem of the DR programs, which is the limitation of consumer’s knowledge to respond upon receiving DR signals. To endorse productiveness and effectiveness of the proposed WBFA-based strategy, substantial simulations are carried out. Furthermore, the proposed WBFA-based strategy is compared with benchmark strategies including binary particle swarm optimization (BPSO) algorithm, genetic algorithm (GA), genetic wind driven optimization (GWDO) algorithm, and genetic binary particle swarm optimization (GBPSO) algorithm in terms of energy consumption, cost of electricity, PAR, and UC. Simulation results show that the proposed WBFA-based strategy outperforms the benchmark strategies in terms of performance metrics.

## 1. Introduction

With the rapid growth in population and economic development, dependence on electrical energy is ever so increasing and, consequently, the energy consumption is on the hike. To further emphasize, the authors recorded that the electricity demand of the energy sector will increase to 40% and both the commercial and residential sectors will increase to 25% by 2025 [1]. The obsolete electric grid is not capable of coping with this rising electricity demand and contemporary challenges like hybrid generation, two-way communication, and two-way power flow. Therefore, the modern power grid, namely smart grid (SG), developed as intelligent electric grid that accommodates internet-of-things (IoT), modern control technologies, information and communication technologies (ICTs), two-way power flow, and hybrid generation. In order to cope with this rising electricity demand, SGs may actively involve either of the two programs: installation of power generating plants or broadcast demand response (DR) programs for energy management [2].

DR programs are the key incentive programs of the SGs that persuade consumers to participate in the electricity market via advanced metering infrastructure (AMI). The DR programs have two classes: (a) incentive-based DR programs and (b) price-based DR programs. In (a), distribution system operators (DSOs) are IoT-enabled agents that remotely control consumer’s appliances when needed with short notice beforehand. In (b), IoT-enabled users are stimulated to spontaneously manage their power usage in response to offered price-based incentives [3]. Since residential buildings have an energy consumption of more than 80%—a remarkable portion of the total energy—(b) is an imperative program that produces affirmative results for both DSOs and consumers while performing energy management [4].

In DR programs, one challenge is the lack of user knowledge which prevents users from participation [5]. One of the developed solutions is to employ automatic controllers in users’ premises that actively participate and contribute to solving an optimization problem, known as energy management controller (EMC). The EMC when enabled with IoT will effectively minimize consumer’s cost of electricity without sacrificing UC, which is a motivation for end-users to take part in DR programs [6]. The EMC output is the optimal power usage schedule of residential building smart appliances. Besides, smart appliances, plugin hybrid electric vehicles (PHEVs), renewable energy sources (RESs), and energy storage systems may penetrate to residential buildings in order to improve sustainability [7,8]. Thus, in-home PHEVs and energy storage systems facilitate consumers to store energy from RESs during daytime and discharge during nighttime to return many benefits from the investment. However, objectives are achieved at high capital cost. The authors in References [9,10,11] proposed power usage scheduling strategies for residential buildings energy management. The developed strategies are affective in minimizing cost of electricity as well as peak electricity demand. Moreover, in these works, users are attracted to active participation due to cost minimization without sacrificing UC. The authors introduced a novel concept of user priority in energy management systems via power usage scheduling using DR programs [12,13,14,15,16]. Home appliances with priority as well as thermal and operational constraints enable EMC to turn on and off appliances on a priority basis.

The above literature provides enough studies relevant to the theme of efficient energy management in SG. Though the focus of some studies is on minimization of cost, some studies catered peak demand reduction, some studies catered alleviation peak-to-average ratio (PAR), and some studies handled UC. To the best of our knowledge, none of the aforementioned studies fully utilized AMI, DR programs, and IoT-enabled environments of the SG to satisfy users and DSOs both parties at the same time. Therefore, in this study, we utilize AMI, DR programs, and IoT-enabled environments of the SG to perform efficient energy management of residential buildings in smart cities in order to minimize cost, curtail PAR, and maximize UC, simultaneously for both users and DSOs satisfaction. The highlights and distinguishing features of this study are given below:A practical optimization model is formulated for efficient energy management of residential building by power usage scheduling of IoT-enabled smart appliances utilizing AMI and different DR programs like time of use pricing scheme (ToUPS), day-ahead pricing scheme (DAPS), and real-time pricing scheme (RTPS) of the SG.The most popular DR programs like ToUPS, DAPS, and RTPS are mathematically modeled and implemented for efficient energy management of residential buildings in smart cities.In References [16,17,18,19], minimization of electricity cost, and alleviation of PAR objectives are catered. In this work, in addition to electricity cost and PAR, the UC in terms of waiting time or delay is formulated and investigated to solve the energy management problem by power usage scheduling of residential building smart appliances using DR programs in IoT-enabled environment of the SG.Optimization problem and constraints are constructed for managing power usage of IoT-enabled residential building smart appliances via scheduling to reduce cost of electricity, alleviate PAR, and maximize UC.A wind-driven bacterial foraging algorithm (WBFA) is developed for IoT-enabled EMC to actively participate in price-based DR programs in order to return optimal power usage schedule for residential building smart appliances.The efficacy of the proposed WBFA-based strategy is validated by comparing it to the benchmark strategies based on GA [15,16], BPSO algorithm [17], GBPSO algorithm [18], and GWDO algorithm [19] in terms of objectives.

The rest of this manuscript is arranged in this manner: First, related work is discussed in Section 2. In Section 3, the proposed energy management framework is discussed. Energy management via scheduling problem description and formulation are described in Section 4. Proposed and benchmark strategies are described in Section 5. Extensive simulations are conducted and their results are discussed in Section 6. At last, in Section 7, the manuscript is concluded and also research directions are provided as future work.

## 2. Related Work

In SG, in the field of energy management, a lot of literature work has been carried out to cope with the rising electricity demand. The literature work relevant to the theme is classified into types: (a) energy management based on mathematical models, (b) energy management based on meta-heuristic and heuristic methods, and (c) energy management based on hybrid methods. This classification is for better understanding. The detailed demonstration is as follows:

### 2.1. Energy Management Based on Mathematical Models

In Reference [20], the authors developed linear programming (LP) for scheduling battery charging/discharging and smart appliances operation using DAPS and RTPS DR programs for the purpose to facilitate consumers in terms of reduction of electricity expenses and maximization of UC. An integer LP (ILP)-based energy management system mechanism is developed in Reference [21] to reduce cost and mitigate peak load. The developed model is a hybrid architecture of PV and power grid serving residential buildings load. Although, the desired objectives are obtained at the expense of increased complexity and high execution time. A novel mixed-integer non-LP (MINLP)-based residential load scheduling mechanism is developed for efficient energy management using RTPS [22]. An MINLP is implemented in Reference [23] for energy management using automated DR programs. The residential buildings appliances such as thermal, critical, non-deferrable, and deferrable are scheduled to minimize cost of electricity with acceptable UC. Novel fractional programming tools for home energy management are developed in Reference [24]. The aim is to minimize cost by household load scheduling under distributed energy resources (DERs) and electric utility companies. The authors developed in Reference [25] MINLP-based prototype for scheduling heating, ventilating, and air-conditioning (HVAC) systems using cost and HVAC constraints. A predictive mixed integer programming (PMIP)-based scheduling mechanism is developed for residential building energy management [26]. The authors aim to minimize cost of electricity and to alleviate peaks in demand. The authors in Reference [27] proposed a novel mechanism based on stochastic model predictive control (SMPC) and MINLP for household appliances and energy resource scheduling. A model predictive control (MPC) algorithm-based collaborative economic model for power usage scheduling of smart cities is developed in Reference [28]. The combined approach aims to cope with rising electricity demand with available energy as well as to minimize cost. An intelligent MILP-based model is developed for the urban area having diverse energy sources [29]. The authors aim to raise their monetary income due to cost reduction. The related work of energy management based on mathematical methods is summarized in Table 1.

### 2.2. Energy Management Based on Meta-Heuristic and Heuristic Methods

The authors proposed a scheduling strategy based on heuristic algorithms like a glowworm swarm particle optimization algorithm for efficient energy management in Reference [30]. The model developed is endorsed by comparative evaluation with the benchmark schemes in terms of performance metrics. The novel mechanism is composed of machine learning models and heuristic algorithms for household load scheduling like fixed load, deferrable load, and regulate-able load using the DR program [31]. A framework based on an evolutionary algorithm (EA) is developed in Reference [32] for demand-side management (DSM) of commercial, residential, and industrial sectors. The authors aim to minimize cost and peaks of electricity consumption to cope with rising electricity demand. However, the cost of electricity minimization in the residential sector is less as compared to both the industrial and commercial sectors. Authors developed a strategy based on bacterial foraging optimization (BFO) algorithm to minimize cost of electricity and discomfort of users [33,34,35,36].

The authors employed GA-based EMC for household appliances scheduling in References [37,38,39]. The GA-based EMC performs household appliances scheduling using combined RTPS and inclined block rate scheme (IBRS) in order to minimize cost of electricity, mitigate peak load demand, and to solve energy scarcity problem.

The EMC based on the particle swarm optimization (PSO) algorithm and its variant are employed in Reference [40,41,42,43] for scheduling energy resources, residential building smart appliances and battery charging/discharging scheduling in order to meet rising electricity demand. This study aims to mitigate peak load demand and cost of electricity simultaneously. A novel WDO algorithm is developed for solving household appliances scheduling in References [44,45]. The EMCs employed based on the WDO algorithm and its variants are for the purpose to minimize the cost of electricity and UC in terms of waiting time. Meta-heuristic algorithms are used to program EMC for residential building power usage scheduling in References [46,47]. These models are beneficial because it minimizes cost of electricity and PAR, which is beneficial for users as well as DSOs. The related work of heuristic and meta-heuristic algorithms is presented in Table 2.

### 2.3. Energy Management Based on Hybrid Methods

An intelligent energy management strategy is developed based on a hybrid of the teacher learning algorithm and GA using a DR program to schedule household load in order to minimize cost of electricity, alleviate PAR, and maximize UC in Reference [48]. However, the objectives are obtained at the expense of sacrificing system simplicity. The authors cascaded the BPSO algorithm with ILP to schedule thermostatically controlled, non-interruptible, and interruptible household load using price-based DR programs [49]. The major goal of the authors is to reduce cost as well as thermal discomfort. Although, the objective function and constraints for solving energy management problems seem unpractical. An energy management strategy is developed using hybrid ToUPS and IBRS for scheduling controllable and uncontrollable household appliances subjected to priority and operational constraints in Reference [50]. This work aims to minimize cost of electricity and to maximize UC. However, cost of electricity is reduced at the expense of increased PAR, which disturbs the stability of the power system. A framework is developed in Reference [51], which is based on a hybrid genetic gravitational search (HGGS) algorithm. The proposed framework is based on HGSA schedule household load in a cloud computing environment to reduce the aggregated cost of electricity. The proposed strategy has outstanding performance as compared to the individual PSO and gravitational search (GS) algorithms. However, PAR and UC are ignored though they are directly related to the bill of electricity. The related work of energy management based on hybrid methods is summarized in Table 2.

## 3. Proposed Energy Management Framework

The proposed framework is elaborated in this section. The DSOs are IoT-enabled agents that transfer DR programs like ToUPS, DAPS, and RTPS to stimulate IoT-enabled users to actively participate in energy management via power usage scheduling of residential building smart appliances using received DR signals. The proposed framework composed of IoT-enabled DSOs and residential buildings utilizing AMI of the SG. The residential building is enabled with IoT and equipped with EMC, home gateway, smart appliances, smart meter, remote control, indoor display (IDD), and wireless home area network. The schematic diagram of the proposed framework is depicted in Figure 1.

The AMI is an essential attribute of the IoT-enabled SG that performs a pivotal role in the central nervous system in the area of energy management via power usage scheduling. The AMI is a two-way communication infrastructure between IoT-enabled DSOs and consumers. The key role of AMI is to collect and deliver recorded energy consumption from smart meters to the DSOs and to transmit DR pricing signals from DSOs to consumers via smart meters and residential building gateway in real-time. The residential building gateway could be separate equipment or might be an integrated entity with smart meter that establish an interface between HAN and wired network. The smart meter could be an indoor or outdoor entity installed in residential buildings between EMC and AMI. The key responsibility of smart meter is to measure, record, and process consumed energy data, and delivered it to the DSOs. Furthermore, it sends DR pricing signals to the IoT-enabled EMC to perform efficient energy management.

In this study, the residential building under consideration is equipped with smart appliances like power adjustable appliances, critical appliances, and time adjustable appliances. The power adjustable appliances have elastic rated power and follow a pre-defined operating schedule. The time adjustable appliances have an elastic operational time and are designed to operate with fixed power rating. They are further classified into two classes like interruptible (dishwasher, tumble dryer, and washing machine) and non-interruptible (electric water heater and vacuum cleaner) [18]. The IoT-enabled EMC is employed in the residential buildings, which is programmed with our proposed WBFA to respond in real-time to receive DR pricing signals to combat the limitation of the dearth of users’ knowledge, which is the hurdle that prevents implementation of DR programs. The employed WBFA-based EMC enabled with IoT takes smart appliances power rating, price-based DR programs, length of time operation, and accessible power grid energy as inputs in order to schedule power usage of residential building smart appliances in the presence of objective function and constraints. The IoT-enabled EMC in a residential building can communicate with smart appliances through diverse communication links like Wi-Fi, ZigBee, HomePlug, and Z-Wave in order to share the generated power usage schedule with smart appliances. The energy management process of the residential building via power usage scheduling of smart appliances is monitored through IDD or remotely through laptops or mobile phones using the IoT facility. All of the processes is illustrated through the working flow in Figure 2.

The proposed energy management framework aims to remotely control and monitor the power usage of residential building smart appliances in order to manage energy by scheduling without human intervention using DR programs. The major objectives of the proposed energy management framework are given below:Electricity cost minimizationPAR alleviationUC maximizationEffective energy utilization

These objectives are achieved by employing WBFA-based EMC that schedules power usage of residential building smart appliances using price-based DR programs by effective energy utilization in the IoT-enabled environment in the SG.

### 3.1. Proposed Framework Inputs

Inputs to the proposed framework are available energy from power grid, DR programs, appliances power rating, length of time operation, and power usage pattern. The detailed demonstration of the inputs are given below.

#### 3.1.1. Residential Building Smart Appliances

The residential building (smart home) is equipped with smart appliances like power adjustable appliances $A_{a}^{P}$, time adjustable appliances $A_{a}^{T}$, and critical appliances $A_{a}^{C}$. The residential building smart appliances have the following parameters that are clearly defined operational time interval, power rating, priority, category, status, and position. The mathematical description is as follows:(1)$$A=\left\{A_{a}^{T},A_{a}^{P},A_{a}^{C}\right\}$$

The status indicator $S_{t}^{a}=\left\{1,0\right\}$ and position indicator $X_{t}^{a}=\left(r_{t}^{a},w_{t}^{a}\right)$ for every smart appliance *a* at timeslot *t* are defined, where $r_{t}^{a}$ represents an appliance remaining timeslots, and $w_{t}^{a}$ represents an appliance waiting timeslots. The detailed demonstration is given below:Power adjustable appliances: These types of smart appliances have elastic rated power and operate with min. rated power during high-price timeslots and operate with max. rated power during low-price timeslots to participate in cost minimization, alleviation of PAR, and maximization of UC. A second priority is assigned to such types of appliances. These smart appliances are represented by $A_{a}^{P}$. These appliances are also termed as power-regulating appliances. Power-adjustable appliances positioned at the initial and next timeslots are formulated as follows:
(2)$$X_{t}^{N}=\left(T_{P}^{o},\beta -\alpha +T_{P}^{o}+1\right)$$
(3)$$X_{t+1}^{P}=\left\{\begin{array}{c} r_{t}^{P},\;0,\;P_{r}^{\min p}\;\;\mathrm{if}S_{t}=1,r_{t}^{P}\geq 1\\ r_{t}^{P},\;0,\;P_{r}^{\max p}\;\;\mathrm{if}S_{t}=1,r_{t}^{P}\geq 1\\ 0,\;0\;\;\;\;\mathrm{otherwise} \end{array}\right.$$
where $X_{t}^{P}$ and $X_{t+1}^{P}$ denote current status and next timeslot status of power-adjustable appliances, respectively. The parameter $T_{P}^{o}$ is total operation timeslots, α is operation start time, β is operation end time, $r_{t}^{P}$ is the number of remaining timeslots, and $S_{t}$ is the status indicator for power adjustable appliances. The power adjustable appliances regulate between minimum power rating $P_{r}^{\min p}$ and maximum power $P_{r}^{\max p}$.Time adjustable appliances: These smart appliances have time elastic operating mechanisms and operate with rated power. These appliances are represented by $A_{a}^{T}$. These appliances are further categorized as: non-interruptible time adjustable appliances $A_{T}^{NI}$ and interruptible time adjustable appliances $A_{T}^{I}$. The third and fourth priorities are assigned to such type of appliances, respectively. The mathematical definition is as follows:
(4)$$A_{a}^{T}=\left\{A_{T}^{I},A_{T}^{NI}\right\}$$Interruptible time adjustable appliances $A_{T}^{I}$: Operation time shifting like advance or delay of such types of appliances is permitted. These appliance operations can be interrupted and delayed even during run time before to finish the assigned task if needed. Such a type of appliance highly contributes in minimization of electricity. Furthermore, these smart appliances could not be turned on during high-price hours and could be shutdown or shifted to low-price hours in order to ensure cost minimization. These smart appliances are also termed deferrable appliances. The position of such types of appliances for current and next timeslots is defined as follows:
(5)$$X_{t}^{I}=\left(T_{I}^{o},\beta -\alpha +T_{I}^{o}+1\right)$$
(6)$$X_{t+1}^{I}=\left\{\begin{array}{c} r_{t}^{I},\;w_{t}^{I}-1,\;P_{r}^{I}\;\mathrm{if}S_{t}=0,w_{t}^{I}\geq 1\\ r_{t}^{I}-1,\;w_{t}^{I}\;P_{r}^{I}\;\mathrm{if}S_{t}=1,r_{t}^{I}\geq 1 \end{array}\right.$$
where $X_{t}^{I}$ and $X_{t+1}^{I}$ denote current status and next timeslot status of interruptible time adjustable appliances, respectively. The parameter $T_{I}^{o}$ is total operation timeslots, α is operation start time, β is operation end time, $r_{t}^{I}$ is the number of remaining timeslots, $w_{t}^{I}$ is the number of waiting timeslots, $P_{r}^{I}$ is the power rating, and $S_{t}$ shows on/off status of smart appliances and of interruptible time adjustable appliances.Non-interruptible time adjustable appliances $A_{T}^{NI}$: Non-interruptible time adjustable appliances are delay tolerable and does not tolerate interruption during operation until the completion of task. The position of non-interruptible time adjustable appliances at current and next timeslots is defined as follows:
(7)$$X_{t}^{N}=\left(T_{N}^{o},\beta -\alpha +T_{N}^{o}+1\right)$$
(8)$$X_{t+1}^{N}=\left\{\begin{array}{c} r_{t}^{N},\;w_{t}^{N}-1,\;P_{r}^{NI}\;\mathrm{if}S_{t}=0,w_{t}^{N}\geq 1\\ r_{t}^{N}-1,\;0\;,\;P_{r}^{NI}\;\mathrm{if}S_{t}=1,r_{t}^{N}\geq 1 \end{array}\right.$$
where $X_{t}^{N}$ and $X_{t+1}^{N}$ denote current status and next timeslot status of non-interruptible time adjustable appliances, respectively. The parameter $T_{N}^{o}$ is total operation timeslots, α is operation start time, β is operation end time, $r_{t}^{N}$ is the number of remaining timeslots, $w_{t}^{N}$ is the number of waiting timeslots, $P_{r}^{NI}$ is power rating, and $S_{t}$ shows on/off status for these appliances.Critical appliances: These smart appliances operate at rated power and cannot tolerate delays or interruption once the operation has started. These appliances serve on a priority basis. These appliances have a predefined schedule that does not disturb UC. These appliances are represented by $A_{a}^{C}$.

Smart appliance input parameters equipped with the residential building are briefly described and presented in Table 3.

#### 3.1.2. ToUPS, DAPS, and RTPS Price-Based DR Programs

This study introduces ToUPS, DAPS, and RTPS price-based DR programs that are fed as an input to our proposed WBFA-based EMC in the IoT-enabled environment. These DR pricing signals are delivered by DSOs to the WBFA-based EMC to schedule the power usage of smart appliances in order to minimize the cost of electricity, alleviate PAR, and maximize UC. These price-based DR programs are adopted from federal energy regulating commission (FERC) [52] as shown in Figure 3. The DAPS price-based DR program is commonly used for household load scheduling of IoT-enabled smart cities. In DAPS, price-based DR programs have three price levels for the complete day, like high, medium, and low price hours as shown in Figure 3. In similar passion, RTPS is defined for the whole day with hour resolution as depicted in Figure 3. Similarly, ToUPS price-based DR programs have also three varying price levels: off, mid, and on-peak price hours. The ToUPS is mathematically modeled as follows:(9)$$\gamma \left(t\right)=\left\{\begin{array}{c} \gamma_{1},\;\;\mathrm{if}\;t\in T_{1}\\ \gamma_{2},\;\;\mathrm{if}\;t\in T_{2}\\ \gamma_{3},\;\;\mathrm{if}\;t\in T_{3} \end{array}\right.$$
where γ(t) is the ToUPS electricity price at timestep *t* and $\gamma_{1}$, $\gamma_{2}$, and $\gamma_{3}$ are the prices at off-peak, mid-peak, and on-peak periods, respectively. The ToUPS is defined for the whole day with hour resolution or alternatively $T_{1}\cup T_{2}\cup T_{3}=24\;h$ and $\gamma_{1}<\gamma_{2}<\gamma_{3}$. The ToUPS price-based DR program is also illustrated in Figure 3. The time interval from 1 to 8 and 22 to 24 h are off-peak hours and corresponds to $T_{1}$ and $\gamma_{1}$ of Equation (Equation 9). Similarly, the time intervals from 8 to 16 and 21 to 22 h are mid-peak hours reflecting $T_{2}$ and $\gamma_{2}$ in Equation (Equation 9). Furthermore, the time interval from 16 to 21 h is on-peak hours and indicates $T_{1}$ and $\gamma_{3}$ in Equation (Equation 9). The EMC based on our proposed WBFA shifts the power usage from on-peak hours to off-peak hours in order to minimize cost electricity, to alleviate PAR, and to maximize UC in IoT-enabled environment of the SG.

### 3.2. Proposed Framework Outputs

The WBFA-based EMC utilizes residential building smart appliance parameters (operation time interval, power rating, priority, status, etc.), price-based DR programs, and accessible power grid energy as inputs to solve the optimization problem by scheduling power usage of residential buildings under IoT-enabled environment of the SG. The smart appliances of the residential building follow the schedule assigned by WBFA-based EMC to minimize cost, to alleviate PAR, and to maximize UC by effective energy utilization. The output of the proposed framework is an optimal power usage schedule of smart appliances of a residential building which actively participate in the energy management area of the SG.

## 4. Energy Management Via Scheduling Problem Description and Formulation

First, the energy management problem is stated and formulated for each objective individually like cost of electricity minimization, PAR alleviation, and UC maximization. Then, the overall energy management problem is stated and formulated as an optimization problem. The detailed demonstration is given below.

### 4.1. Energy Management Problem Description

Energy management via power usage scheduling is a cumbersome and challenging problem due to stochastic, nonlinear, and random behavior of end-users. In this context, most researchers focus on energy management via power usage scheduling of residential building smart appliances. Numerous strategies have been proposed in the literature to manage power usage of residential buildings via smart appliance scheduling using price-based DR programs in an IoT-enabled environment of the SG. The authors in References [15,16] developed a GA-based strategy for appliance scheduling in order to reduce the cost of electricity and to mitigate PAR. However, UC is sacrificed while minimizing the cost of electricity. This strategy has some inherent limitations and the problem of unguided mutation that makes loads unbalanced [53]. A BPSO algorithm-based strategy is developed for smart appliance scheduling of residential buildings in Reference [17]. However, the scheduling time horizon further divides into shorter timeslots and increases the model complexity that results in an increased computation overhead. This strategy has the limitation of increased model complexity of which could be avoided.

Thus, keeping in view the above challenges, first, we proposed the WBFA algorithm, and then we developed a strategy, where EMC is programmed with our proposed WBFA algorithm to automatically respond to DR pricing signals to schedule power usage of residential buildings to ensure efficient energy management. We select WDO and BFO algorithms and proposed the WBFA algorithm due to the following characteristics: the ease of implementation, flexibility for specified constraints, low computational complexity, low computational time, and fast converging speed. The energy management problems are solved via power usage scheduling using DR signals. The solution could be the optimal power usage schedule of residential building smart appliances in IoT-enabled environment of the SG. The end-users will follow the returned optimal power usage schedule in order to effectively utilize available energy and to minimize the cost of electricity, mitigate PAR, and to maximize UC. In subsequent sections, each objective is demonstrated and formally formulated as follows:

### 4.2. Residential Building Energy Consumption Formulation

The power consumption is the consumed electrical energy of smart appliances of the residential building during the scheduling period. This study assumed that a residential building has three types of smart appliances like power adjustable appliances $A_{a}^{P}$, critical appliances $A_{a}^{C}$, and time adjustable appliances $A_{a}^{T}$.

First, time adjustable appliances have two more types: (a) interruptible time adjustable, and (b) non-interruptible time adjustable appliances. The consumed electricity per timeslot of interruptible appliances is formulated as follows:(10)$$E_{c}^{I}\left(t\right)=P_{r}^{I}\times S_{t}$$
where $E_{c}^{I}\left(t\right)$ is the consumed electricity per timeslot, $P_{r}^{I}$ represents power rating, and $S_{t}$ is the on/off status indicator of (a) catagory appliances. The net consumed electricity by (a) category appliances is formulated as follows:(11)$$E_{T}^{I}=\sum_{t=1}^{24}\sum_{a=1}^{N}E_{c}^{I}\left(t\right)\;\forall \;I\in A$$

The consumed electricity of (b) category appliances at each timeslot is formulated as follows:(12)$$E_{c}^{NI}\left(t\right)=P_{r}^{NI}\times S_{t}$$
where $E_{c}^{NI}\left(t\right)$ is the consumed electricity at each timeslot *t*, $P_{r}^{NI}$ represents power rating, and $S_{t}$ is the on/off status indicator of (b) catagory appliances. Thus, net the consumed electricity by (b) category appliances is formulated as follows:(13)$$E_{T}^{NI}=\sum_{t=1}^{24}\sum_{a=1}^{N}E_{c}^{NI}\left(t\right)\;\forall \;N\in A$$

Thus, the net consumed electricity per day by time adjustable appliances can be formulated as follows:(14)$$E_{T}^{ta}=E_{T}^{I}+E_{T}^{NI}$$
where $E_{T}^{I}$ and $E_{T}^{NI}$ are the per day consumed electricity by (a) category appliances and (b) category appliances, respectively, and $E_{T}^{ta}$ represents the net consumed electricity of both (a) and (b) categories appliances.

The consumed electricity per timeslot and per day of power adjustable appliance then can be formulated as follows:(15)$$E_{c}^{p}\left(t\right)=\left\{\begin{array}{c} P_{r}^{\min p}\times S_{t}\;\;\mathrm{for}\;\mathrm{on}-\mathrm{peak}\;\mathrm{timeslots}\;\mathrm{of}\;\partial \left(t\right),\;\rho \left(t\right),\;\gamma \left(t\right)\;\;\\ P_{r}^{\max p}\times S_{t}\;\;\mathrm{for}\;\mathrm{off}-\mathrm{peak}\;\mathrm{timeslots}\;\mathrm{of}\;\partial \left(t\right),\;\rho \left(t\right),\;\gamma \left(t\right)\;\; \end{array}\right.\;\forall \;p\in A$$
(16)$$E_{T}^{p}=\sum_{t=1}^{24}\sum_{a=1}^{N}E_{c}^{p}\left(t\right)$$

### 4.3. Cost Formulation for Consumed Electricity

Electricity cost is the dues to be deposited by the users to DSOs for using electrical energy for a specified time horizon. This study formulated cost of electricity using ToUPS, DAPS, and RTPS price-based DR programs offered by DSOs. The FERC in 2009 noticed that users who actively participated in DR programs and shifted their load from on-peak hours to off-peak hours have received 65 percent benefit. The cost paid by users to DSOs for consumed electricity using DAPS can be formulated as follows:(17)$$C_{T}^{DA}=\sum_{t=1}^{24}\left(\sum_{a=1}^{N}E_{c}^{a}\left(t\right)\times S_{t}\times \partial (t)\right)$$
Equation (Equation 17) denotes cost to be deposited by consumers for consumed electricity using DAPS price-based DR program. The $C_{T}^{DA}$ is the net cost paid by users for operating all categories of residential building smart appliances, $E_{c}^{a}\left(t\right)$ is consumed electricity of each appliance *a* at timeslot *t*, and ∂(t) is DAPS for each timeslot *t*.

Similarly, cost to be paid for operating residential building smart appliances using RTPS and ToUPS can be formulated as follows, respectively:(18)$$C_{T}^{RP}=\sum_{t=1}^{24}\left(\sum_{a=1}^{N}E_{c}^{a}\left(t\right)\times S_{t}\times \rho \left(t\right)\right),$$
(19)$$C_{T}^{UP}=\sum_{t=1}^{24}\left(\sum_{a=1}^{N}E_{c}^{a}\left(t\right)\times S_{t}\times \gamma \left(t\right)\right),$$
where $C_{T}^{RP}$ and $C_{T}^{UP}$ denote net electricity bill to be paid under RTPS and ToUPS, respectively. Furthermore, the parameters ρ(t) and γ(t) represent the offered price rates to the consumers through RTPS and ToUPS, respectively.

### 4.4. Peak-to-Average Ratio Formulation

The DSOs stimulate users to shift their power usage from on-peak timeslots to off-peak timeslots to alleviate peak load for the purpose of minimizing peaks in demand. PAR is defined as the ratio of peak power usage to average power usage. It is imperative for both DSOs and users due to two reasons: (a) smoothen out the load to avoid the need for peak power plants and (b) minimizes users’ cost of consumed electricity. The PAR is formulated as follows:(20)$$R_{a}^{p}=24\times \left(\frac{\max\left(E_{c}^{I}\left(t\right),\;E_{c}^{NI}\left(t\right),\;E_{c}^{p}\left(t\right),\;E_{c}^{c}\left(t\right)\right)}{E_{T}}\right),$$
where $R_{a}^{p}$ represents the PAR, and $E_{T}$ denotes total energy consumption.

### 4.5. User Comfort and Discomfort

UC is related to energy consumption, waiting time, temperature, illumination, air quality, humidity, sound, and the demographic profile of the residents [54,55]. In this work, the UC in terms of waiting (how much delay a user faces for activity by shifting appliances from on-peak hours to off-peak hours) is considered. The optimal power usage before and after scheduling is different because the operation of appliances is shifted from high-price timeslots to low-price timeslots. Therefore, the user confronts frustration that is formulated in respect of waiting time. As the trades-off exists between the cost of electricity and waiting time, the users who could wait more would deposit less cost and the users who could not wait more would deposit high cost. The waiting time of power-adjustable appliances is zero due to predefined 24-h lengths of operation time. Such types of appliances contribute to scheduling through their power flexible nature. The UC in terms of waiting time is formulated as follows:(21)$$w_{a}=\frac{\sum_{t=1}^{T}\sum_{a=1}^{n}\left|\left(T_{a,t}^{o,unsch}-T_{a,t}^{o,sch}\right)\right|}{T_{a}^{lo}}$$
where $w_{a}$ represents waiting that each appliance *a* may face due to appliance shifting, $T_{a,t}^{o,unsch}$ is operation status of appliances before scheduling, $T_{a,t}^{o,sch}$ is operation status of appliances after scheduling, and $T_{a}^{lo}$ is the appliances total length of operation time. The WBFA-based EMC adjusts and shifts the appliances before and after the specified time in response to price-based DR programs and consumer’s priority. The maximum delay (waiting time) that an appliance can bear is determined as follows:(22)$$w_{a}^{d}=T_{a}^{t}-T_{a}^{lo}$$
where $w_{a}^{d}$ is the maximum delay that an appliance may face while shifting operation from on-peak hours to off-peak hours and $T_{a}^{t}$ is the appliances total time interval. UC is sacrificed with the increase in $w_{a}^{d}$. The user discomfort is maximum when $w_{a}$ is equal to $w_{a}^{d}$; this is the worst case, which might not happen usually. The user discomfort in respects of percentage is formulated as follows:(23)$$D=\frac{w_{a}}{w_{a}^{d}}\times 100$$

### 4.6. Overall Energy Management Optimization Problem Formulation

This study formulates the power usage scheduling of residential buildings for efficient energy management as an optimization problem. It is preferable for an optimization perspective that IoT-enabled residential buildings effectively utilize available energy in such a way to maximize UC, mitigate PAR, and minimize the cost of electricity paid to DSOs. The energy management problem is formulated as an optimization problem as follows:(24)$$\min\;\left(C_{T},w_{a},R_{a}^{p}\right)$$
subject to
(25)$$E_{T}\leq Capacity$$
(26)$$E_{T}^{sch}=E_{T}^{unsch}$$
(27)$$T_{i}^{o,sch}\neq T_{i}^{o,unsch}$$
(28)$$T_{a}^{lo,sch}=T_{a}^{lo,unsch}$$

The constraint in Equation (Equation 25) represents the power grid capacity constraint that confirms that the power grid is not overburdened and is capable to take part in power usage scheduling of residential buildings. Through the constraint in Equation (Equation 26), it is ensured that the net energy consumption before and after scheduling is equal. The constraint in Equation (Equation 27) denotes the status of activity whether it is in-progress or completed. This constraint is in support of fair comparison. Equation (Equation 28) ensures that length of time interval before and after scheduling must be same, subject to a fair comparison.

## 5. Proposed and Benchmark Strategies for Efficient Energy Management

The WBFA-based strategy and benchmark strategies like BPSO algorithm, BFO algorithm, WDO algorithm, GA, GWDO algorithm, and GBPSO algorithm are adopted to solve the optimization problem of energy management of residential buildings by optimal power usage scheduling. The detailed description of the proposed and benchmark strategies are as follows:

### 5.1. GA-Based Strategy for Efficient Energy Management

In References [15,16], GA based strategy is adopted to schedule residential building smart appliances for the purpose of minimizing PAR and the cost of electricity. The DR program, accessible power grid energy, and residential building smart appliance operational parameters are fed as input to the GA. The GA uses these inputs to generate a solution of the population for input chromosomes. Each chromosome represents a candidate solution. The GA-based strategy evaluates the fitness function using randomly created population and stores the best results returned during evaluation. The stored best solution is passed through the crossover and mutation phase to acquire the global optimal solution. The crossover probability and mutation probability parameters are control parameters of GA, which are directly related to the convergence rate. In order to obtain optimal power usage scheduling, the crossover rate is fixed at cr=0.9 and the mutation rate is fixed at pm=0.1. The population returned after crossover and mutation phase will be the optimal population representing the optimal power usage schedule of residential building smart appliances. The controlling parameters of GA are listed in Table 4. The stepwise procedure of GA is shown in Figure 4 for a better understanding of the GA-based strategy for IoT-enabled residential building smart appliances.

### 5.2. BFO Algorithm-Based Strategy for Efficient Energy Management

The BFO algorithm-based strategy [56] is elaborated in this section. The BFO algorithm is stimulated through the foraging manners of a bacteriu1m. The bacterium swims in search of nutrients to discover the best nutrients for the purpose to maximize energy. In optimization problems, the fittest nutrients represent the optimal solution. The BFO algorithm-based strategy comprises of three steps: (i) elimination-dispersion, (ii) reproduction, and (iii) chemotaxes. The control parameters of the BFO algorithm-based energy management strategy are listed in Table 4. Before starting the three-step procedure, the first relevant parameters initialization is conducted. Then, the chemotaxes phase starts; in this phase, we randomly generate a bacteria population matrix. In optimization problems, each bacterium position represents a candidate solution. The chemotaxes phase also represents the tumbling or swimming of a bacterium, which is analogous to convergence or divergence of the optimization problem. As discussed earlier, we generated a population matrix randomly generated; that is why initially the solution diverges. The matrix of the population achieved after fitness evaluation is the local optimal population converging towards the local best solution. The stepsize and bacterium position is denoted by θ(i,:) and $c_{i}$ respectively. The solution convergence rate can be controlled with stepsize. Furthermore, with small stepsize solution trapped into local minima and with large stepsize solution diverging from the global optima, the population obtained after chemotaxes are the local best population representing the local best solution. Through the reproduction phase, the search to find the optimal solution is accelerated and refined in order to obtain a feasible solution. The elimination-dispersion step of the BFO algorithm tries to achieve a global optimal solution by discarding irrelevant and redundant information. The global optimal solution obtained represents the optimal power usage schedule of residential building smart appliances, which is utilized for efficient energy management. The stepwise procedure of the BFO algorithm-based strategy is shown in Figure 5.

### 5.3. BPSO Algorithm-Based Strategy for Efficient Energy Management

The BPSO algorithm-based strategy [57] is elaborated in this section. The BPSO algorithm is a binary variant of the PSO algorithm. It works on the principle of birds flock foraging. The birds flock moving in search of food that has a specific position and velocity. The BPSO algorithm-based strategy composed of two steps: (a) position and (b) velocity. The population matrix is controlled by the velocity of the particle, and the candidate solution is controlled by the position of the particle. The particle initial velocity is determined as follows:(29)$$\begin{array}{c} v_{j}=v_{max}\times 2\times \left(rand\left(swarm,n\right)-0.5\right) \end{array}$$

The position matrix is a solution matrix representing an optimal power usage schedule of residential building smart appliances, where each entry in the position matrix represents on/off status of smart appliances. First, we randomly generate the position matrix, and the position matrix is updated via fitness evaluation. After fitness evaluation, the local best solution is acquired known as pbest, and when the termination criterion is met, the global optimal solution is acquired known as gbest. The gbest is the optimal power usage schedule of residential building smart appliances obtained from the BPSO algorithm-based strategy is utilized for efficient energy management. The velocity function includes key parameters like the inertia factor, previous acceleration coefficients, and best position values. These parameters play a vital role in controlling the convergence behavior of the solution. The parameters used in simulations are listed in Table 4. As the BPSO algorithm is a binary variant of PSO algorithm, we used the sigmoidal function with velocity function to obtain the binary variant. The conversion is mathematically stated as follows:(30)$$Sigmod\left(j,i\right)=\frac{1}{1+e^{-v_{new}}}$$

The binary variant of the position matrix is obtained as follows:(31)$$\begin{array}{cc} x_{new} & =1\;if\;rand(1)\leq Sigmoid(j,i)\\ x_{new} & =0\;if\;rand(1)>Sigmoid(j,i) \end{array}$$

The binary variant of the position matrix is fed to the fitness evaluation phase; the evaluation process iterates for several epochs in order to achieve the global best gbest position matrix. The returned global optimal solution represents the optimal power usage schedule of residential building smart appliances, which is utilized for efficient energy management in order to minimize cost of electricity, to alleviate PAR, and to maximize UC. The stepwise procedure of the BPSO algorithm-based strategy for efficient energy management is shown in Figure 6.

### 5.4. WDO Algorithm-Based Strategy for Efficient Energy Management

The WDO algorithm is elaborated in [58]. A WDO algorithm is a nature-inspired algorithm that works on the principal’s wind atmospheric motion. The WDO algorithm is composed of two steps: air parcel position function and air parcel velocity function. In our scenario, the air parcel matrix represents the solution matrix. In *N*-dimensional space, the moving air parcels are exposed to various friction forces such as frictional force, coriolis force, pressure gradient force, and gravitational force. The parameters used in the velocity function update formula are presented in Table 4.

The control parameters of WDO algorithm are initialized. After initialization randomly generate the parcel position matrix, where each entry in the position matrix represents the status of residential building smart appliances. Then, the initial velocity is determined as follows:(32)$$\begin{array}{c} v_{i}=v_{max}\times 2\times \left(rand\left(populationsize,n\right)-0.5\right) \end{array}$$

Now, the randomly created parcel position matrix is updated through the fitness evaluation function. The updated position matrix represents the optimal power usage schedule of residential building smart appliances. The velocity is also updated. Then, the parcel position matrix is updated as follows:(33)$$x_{new}=x_{cur}+u_{new}\Delta t$$

The velocity lower and upper bound during each epoches are determined as:(34)$$\begin{array}{cc} u_{new} & =u_{max}\;ifu_{new}>u_{max}\\ u_{new} & =-u_{max}\;ifu_{new}<u_{max} \end{array}$$

Furthermore, the position and velocity updated procedure iterates until the termination criterion is met or the optimal solution is returned. The WDO algorithm-based strategy output is a global best solution gbest representing power usage schedule of residential building smart appliances. The entire stepwise WDO algorithm-based strategy is shown in Figure 7.

### 5.5. GWDO Algorithm-Based Strategy for Efficient Energy Management

The GBPSO algorithm is elaborated in [18]. The GBPSO algorithm is a hybrid of the GA and BPSO algorithms. The GBPSO algorithm utilizes the beneficial characteristic of both the GA and BPSO algorithms. The GA is effective in the alleviation of PAR, and the BPSO algorithm is efficient in the reduction of electricity cost. Thus, GBPSO algorithm-based strategy fully utilizes the desired features of both algorithms in order to achieve optimal power usage schedule of residential building smart appliance. The purpose of optimal power usage scheduling is to minimize the cost of electricity and PAR, to maximize UC, and to efficiently utilize DSOs energy. The GBPSO algorithm is composed of two steps: executing complete steps of the BPSO algorithm (refer to Section 5.3) and feeding the output of the BPSO algorithm to crossover and mutation phases of GA (refer to Section 5.1). The power usage schedule returned from the GBPSO algorithm is optimal because the steps of GA are applied to the global optimal solution obtained from the BPSO algorithm instead of random values. The GBPSO algorithm-based strategy parameters are presented in Table 4. The entire stepwise procedure of the GBPSO algorithm-based strategy is shown in Figure 8.

### 5.6. GWDO Algorithm-Based Strategy for Efficient Energy Managment

The GWDO algorithm is elaborated in [19]. Likewise, the GWDO algorithm is a hybrid algorithm obtained by cascading the entire WDO algorithm with the crossover and mutation steps of GA. The GWDO algorithm cascades two algorithms because GA is efficient in PAR minimization and the WDO algorithm is efficient in both cost minimization and UC maximization. The GWDO algorithm-based strategy schedules the power usage pattern of residential building smart appliances to satisfy users and DSOs both parties. The GWDO algorithm is of two steps like in the GBPSO algorithm: (a) entire working procedure of WDO algorithm [19] and (b) crossover and mutation steps of GA [16]. The global best result returned from the WDO algorithm is passed through crossovers and mutation steps of GA to achieve optimal power usage schedule of residential building smart appliances. This optimal power usage schedule is utilized by smart appliances in order to minimize the cost of electricity and PAR with affordable UC. The GWDO algorithm parameters are presented in Table 4. The entire working procedure of the GWDO algorithm is shown in Figure 9.

### 5.7. Proposed Wind-Driven Bacterial Foraging Algorithm-Based Strategy for Efficient Energy Management

The WBFA is an algorithm obtained by cascading WDO and BFO algorithms. The proposed algorithm is designed by adopting the whole working procedure of the WDO algorithm and the elimination-dispersion, chemotaxis, and reproduction operation steps of the BFO algorithm. These two algorithms from the pool of heuristic algorithms are picked up because the WDO outperforms in terms cost minimization and UC maximization, and on the other hand, the BFO algorithm is efficient in terms of minimization of PAR. Therefore, key characteristics of both WDO and BFO algorithms are fused in WBFA to schedule power usage of residential building smart appliances in order to ensure the cost of electricity minimization, the alleviation of PAR, and the maximization of UC. The WBFA is composed of four stages: (a) WDO algorithm, (b) elimination-dispersion, (c) reproduction, and (d) chemotaxis. The control parameters of the WBFA-based strategy are listed in Table 4. The output of the lagging stage is fed as input to the leading stage in order to acquire an optimal power usage schedule of residential building smart appliances. The stepwise procedure of WBFA is illustrated in Figure 10. The outstanding performance of the proposed algorithm is due to deeper layer layout and more control parameters as compared to the existing algorithms. This deep layer layout and more control parameters enable the algorithm to obtain our desired objectives. However, due to the deeper layout and more control parameters, it takes more time to execute as compared to the existing algorithm because trade-off exists in nature.

The proposed algorithm WBFA and adapted algorithms like GBPSO, GWDO, GA, and BPSO evaluation in terms of convergence speed and execution time are shown in Figure 11 and Figure 12, respectively. The good convergence speed and high precision of our proposed WBFA algorithm are due to two reasons: (i) BFO algorithm keeps the diversity of the population and remedies the defect of falling in local optima of WDO algorithm and (ii) the WDO algorithm cures the slow speed of the convergence shortcoming of the BFO algorithm. The moderate execution time (122 s) of the proposed algorithm WBFA is due to the application of key steps that BFO on the optimal result returned from the WDO algorithm. The existing algorithms like GBPSO and GWDO have high execution times of 145 s and 133 s, respectively.

The proposed WBFA algorithm and adapted algorithms like GBPSO, GWDO, GA, and BPSO numerical results in terms of convergence speed, execution time, and complexity are depicted in Table 5. Thus, the proposed algorithm has moderate execution time and complexity and fast convergence speed.

## 6. Simulation Results and Discussion

Extensive simulations are conducted to validate our proposed WBFA-based strategy in terms of energy management using price-based DR programs as depicted in Figure 3a–c, respectively. For simulations, MATALAB^®^ M-file platform is utilized installed on Intel^®^ Core™ i3-3420M CPU @ 2.4 GHz and 6 GB RAM with Windows 10. The DAPS price-based DR program has three price levels: low, medium, and high, as shown in Figure 3a. The high-price hours are also known as on-peak hours. Its duration spans from 11 to 16 h. The medium-price hours are also termed as mid-peak hours. Its duration ranges from 1 to 11 h; likewise, a low-price hour is also named as off-peak hours. Its range spans from 18 to 23 h. The remaining timeslots are assumed as medium-price hours. Likewise, RTPS-based price-based DR program has also three price hours: high-price hours ranging from 7 to 10 timeslots, medium-price hours spanning from 11 to 15 timeslots, and the rest of the timeslots as low-price hours. The ToUPS price-based DR program has also three levels likewise DAPS and RTPS price-based DR programs: high-price hours spanning from 16 to 21 timeslots, medium-price hours, and low-price hours. The proposed WBFA-based strategy is compared with benchmark strategies based on GA [15,16], BPSO algorithm [17], GBPSO algorithm [18], and GWDO algorithm [19] in simulations. These existing strategies are selected due to architecture resembling the proposed WBFA-based strategy. For performance evaluation of the proposed WBFA based strategy, four performance metrics are selected from the pool of metrics: (a) cost of electricity, (b) energy consumption, (c) PAR, and (d) UC.

The power usage of residential building smart appliances is scheduled using price-based DR programs as depicted in Figure 13a–c to minimize the cost of electricity, to alleviate PAR, and to maximize UC by efficient energy utilization. The overall power usage profile before scheduling and after the scheduling of residential building smart appliances is shown in Figure 13. The proposed and benchmark strategies are capable of scheduling power usage of the IoT-enabled residential buildings by employing DAPS price-based DR program as depicted in Figure 13a. Figure 13a depicts that both strategies have shifted load from high-price hours to low-price hours by employing DAPS price-based DR program. However, it is obvious that our proposed WBFA-based scheduled power usage profile is comparatively most favorable. The peak power usage of the proposed WBFA-based strategy and benchmark strategies during high-price hours are 4 kWh, 2.3 kWh, 2.2 kWh, 2.4 kWh, and 6.2 kWh, respectively. Therefore, from Figure 13a, it is obvious that the proposed WBFA-based strategy has comparatively average peak power usage. Which is the evidence that proposed WBFA-based strategy schedules power usage of residential building smart appliances in such a manner to minimize the cost of electricity and to maximize UC. The benchmark strategies only catered PAR and cost of electricity. In contrast, when our proposed WBFA-based strategy is not adopted by the users, then they purchase energy from 2 to 9 timeslots and 13 to 17 timeslots, which are medium- and high-price hours that result in peak energy consumption and overwhelm the power grid. Likewise, power usage scheduling of residential building smart appliances using RTPS of proposed and benchmark strategies and before scheduling scenarios are illustrated in Figure 13b. The peak power usage of the proposed WBFA-based strategy is 6 kWh, BPSO is 2 kWh, GA is 7.1 kWh, GBPSO is 3 kWh, and GWDO is 3.9 kWh as illustrated in Figure 13b. Thus, the proposed WBFA-based strategy has average peak power usage as compared to benchmark strategies. This average peak power consumption of the proposed WBFA-based strategy is due to optimal power usage scheduling of residential building smart appliances keeping in view all objectives. On the other hand, the focus of benchmark strategies is only on cost and PAR. The power usage of residential building smart appliances scheduling using ToUPS price-based DR program for the proposed WBFA based strategy and benchmark strategies is shown in Figure 13c. The proposed WBFA-based strategy has a peak power consumption of 5.2 kWh, which is average peak power consumption when comparatively analyzed. The reason for this optimal performance of our proposed strategy is that it performs energy management via power usage scheduling of residential building smart appliance keeping in view UC. On the other hand, benchmark strategies focus only on the cost of electricity and PAR while ignoring UC.

The cost of electricity per hour analysis of the proposed and benchmark strategies for consumed electricity is shown in Figure 14. Figure 14a clearly demonstrates that both strategies (proposed and benchmark) can minimize cost of electricity by scheduling under DAPS price-based DR program as compared to without power usage scheduling. However, it is obvious that the overall cost of electricity per hour of our proposed WBFA-based strategy is low as compared to benchmark strategies. On the other hand, cost of electricity of our proposed WBFA-based strategy during high-price hours is high because of the consideration of UC. The cost of electricity per hour of the proposed WBFA-based strategy and benchmark strategies using RTPS price-based DR program is shown in Figure 14b. It is obvious that our proposed WBFA-based strategy scheduling is optimal as compared with benchmark strategies, and consequently, users enjoy the lowest per hour cost of electricity. Likewise, in the case of ToUPS price-based DR programs, the lowest cost of electricity per hour of the proposed WBFA-based strategy as compared to benchmark strategies is also shown in Figure 14c.

The aggregated cost of electricity numerical values evaluation for both the proposed WBFA-based strategy and benchmark strategies is illustrated in Figure 15. Aggregated cost per day evaluation of the proposed WBFA-based strategy and benchmark strategies for the DAPS price-based DR program is shown in Figure 15a. The net costs of electricity per day of benchmark strategies like GBPSO, GWDO, GA, BPSO, and our proposed WBFA-based strategy are 36 cents, 35 cents, 42 cents, 37 cents, and 34 cents, respectively. The percent decrements of benchmark strategies like GBPSO, GWDO, GA, BPSO, and our proposed WBFA-based strategy are 23.4%, 25%, 10.6%, 21.2%, and 27.6%, respectively. From numerical results, it is obvious that the WBFA-based strategy has minimized the aggregated cost of electricity per day remarkably as compared to the benchmark strategies and without power usage scheduling. The net cost of electricity per day of the proposed WBFA-based strategy and benchmark strategies under the RTPS price-based DR program is illustrated in Figure 15b. The aggregated costs of electricity of the benchmark strategies like GBPSO, GWDO, GA, BPSO, and our proposed WBFA-based strategy are 10.3 cents, 10.1 cents, 15 cents, 16 cents, and 9.5 cents, respectively. Likewise, the aggregated cost using ToUPS price-based DR program of the proposed WBFA-based strategy and benchmark strategies is shown in Figure 15c. Our proposed WBFA-based strategy outperforms existing strategies (GA, BPSO, GBPSO, and GWDO) in terms of electricity bill reduction. The percent decrement of our proposed strategy is 52.10%, which the higher reduction in the bill as compared to the reductions in the electricity bills of GA, BPSO, GBPSO, and GWDO, which are 12.70%, 4.2%, 48.10%, and 50.20%, respectively. The statistical observations of cost of the proposed WBFA-based strategy and existing strategies using DAPS, RTPS, and ToUPS DR programs are listed in Table 6, Table 7 and Table 8, respectively. It is concluded that the proposed WBFA-based strategy outperforms existing energy management strategies in terms of electricity cost minimization.

Proposed WBFA-based strategy and existing strategies evaluation in terms of PAR using ToUPS, DAPS, and RTPS price-based DR programs is shown in Figure 16. The evaluation of both strategies (proposed and existing) in terms of PAR using DAPS is shown in Figure 16a. The users who have not employed our proposed strategy or existing strategies for scheduling have very high PARs of 5. When users employ existing strategies like GWDO, GBPSO, GA, BPSO, and our proposed WBFA-based strategy, the power usage scheduling leads to reduced PAR of 3.8, 3.6, 3.7, 3.1, and 2.1, respectively. Thus, our proposed WBFA-based strategy minimizes cost as well as alleviates PAR by 58%, which is indispensable for SG reliable operation. Likewise, our proposed WBFA-based strategy curtailed PAR by 54% and 62% with RTPS price-based DR program (refer to Figure 16b) and with ToUPS price-based DR program (refer to Figure 16c), respectively. The percent decrement in PAR of our proposed WBFA-based strategy and existing (GA, BPSO, GWDO, and GPSO) strategies using DAPS, RTPS, and ToUPS price-based DR programs is listed in Table 9, Table 10 and Table 11, respectively.

UC and discomfort analysis of the proposed strategy using price-based DR program is depicted in Figure 17. The frustration faced by consumers in terms of waiting time with the proposed strategy under the DAPS price-based DR program is depicted in Figure 17a. The user discomfort (waiting time) and electricity cost are inversely related, and trade-off exists between waiting and electricity cost. The waiting time of all appliances in the IoT-enabled smart home is numerically visualized in Figure 17a; the waiting times of time adjustable appliances, critical appliances, and power-adjustable appliances are high, medium, and low, respectively. The high waiting time of time-adjustable appliances is due to their delay-tolerant nature, and these appliances contribute to the electricity bill reduction. On the other hand, the waiting time of power-adjustable appliances is low because they are not delay-tolerant and take part in scheduling through their power flexible nature. In a similar manner, the waiting time of all smart appliances in IoT-enabled residential building is numerically visualized in Figure 17b,c using RTPS and ToUPS, respectively.

Observation of average waiting time for the proposed and existing energy management strategies in terms of numerical values is illustrated in Figure 18. The average waiting times of the proposed WBFA, and GA, BPSO algorithm, GWDO algorithm, GBPSO algorithms are 4 h, 3 h, 3.3 h, 3.5 h, and 4.1 h, respectively, using DAPS, as depicted in Figure 18a. In terms of average waiting time, the GA-based strategy is superior to all strategies. The most suitable reason for this behavior is the existence of a trade-off between electricity cost and average waiting time. The average waiting time of our proposed WBFA-based strategy is high because electricity cost and PAR are reduced at the cost of moderate user discomfort. The average waiting time (user discomfort) is reduced at the expense of high electricity bill payment, as shown in Figure 18b. The average waiting times under RTPS of the proposed WBFA, and existing GA, BPSO algorithm, GWDO algorithm, GBPSO algorithms are 4.5 h, 3.2 h, 3.3 h, 3.4 h, and 4.4 h, respectively. In a similar passion, user discomfort is reduced at the expense of high electricity bill payment, as depicted in Figure 18c.

## 7. Conclusions and Future Research Directions

Electricity efficient utilization, optimal electricity consumption, and cost minimization can be acquired by employing DR programs. However, the dearth of knowledge prevents the development and employment of DR programs in consumer premises. Diverse methods like classical, mathematical, heuristic, meta-heuristic, and hybrid are applied to implement and employ DR programs for energy management of the residential building in IoT-enabled environment of the SG. In this work, we propose the WBFA algorithm, which is a hybrid of the WDO and BFO algorithms. The EMC programmed with our proposed WBFA algorithm automatically responds to DR programs to participate in the energy management of the residential building in IoT-enabled environment of the SG. We adopted DAPS, RTPS, and ToUPS price-based DR programs; our proposed WBFA-based strategy schedules power usage of residential building smart appliances under these DR programs. The objective of our proposed WBFA-based strategy is to minimize the cost of electricity, to alleviate PAR, and to maximize UC. Simulation results demonstrate that employing ToUPS DR program for energy management leads to the lowest cost of electricity, PAR, and stable power usage schedule of residential building smart appliances compared to DAPS and RTPS DR programs. Furthermore, our proposed WBFA-based strategy minimized cost of electricity and PAR by 27.6% and 58% by employing the DAPS DR program, by 40% and 54% by employing the RTPS DR program, and by 52.1% and 62% by employing the ToUPS DR program, respectively, as compared to without employing price-based DR programs. Thus, employing ToUPS DR program for energy management leads to providing favorable outcomes for both consumers and DSOs.

This study can be extended into various directions in the future:The energy management via scheduling can be performed through the coordination of residential building smart appliances in the presence of power grid, RE, energy storage systems, and EVs by embedding sensors on the participant. To handle such a coordinated environment, the EMC would be made intelligent and smart by incorporating sensing, communication, and the IoT modules on the traditional EMC. Furthermore, for this coordinated environment, net metering is required where consumers would become prosumers. The prosumers can generate renewable energy and store it into the energy storage system and EVs that have storage batteries. The prosumers sell their generated and stored energy back to the power grid in order to ensure reliable, stable, sustainable, and economical power grid operation. The prosumers enabled with intelligent and smart EMC and net metering features can actively participate in regulated energy markets with price-based DR programs and incentive-based DR in order to facilitate both the power grid and consumers.This work can be extended to fog- and cloud-based energy management via scheduling using the DR program in the SG.An energy management model with a hybrid generation (RE and fossil fuel) system can be extended by considering vehicles to grid parking stations as a controllable load based on game-theory-based optimization algorithm.A MILP-based efficient energy management modular framework may be proposed for both urban and ruler energy systems for performance and sensitivity analysis.An innovative energy management model may be proposed in cloud computing for efficient load scheduling using a hybrid genetic gravitational search algorithm.To perform efficient energy management, an intelligent forecaster or machine learning-based forecaster for price and load is required before to schedule the energy consumption pattern of consumers.In the future, we will engage in some advanced, intelligent, and hybrid appliances that have time as well as power-flexible nature in the energy management strategy. Such types of appliances will provide more opportunities for EMC to engage them in energy management in order to provide economical and sustainable solutions.

## Figures and Tables

**Figure 1 sensors-20-03155-f001:**
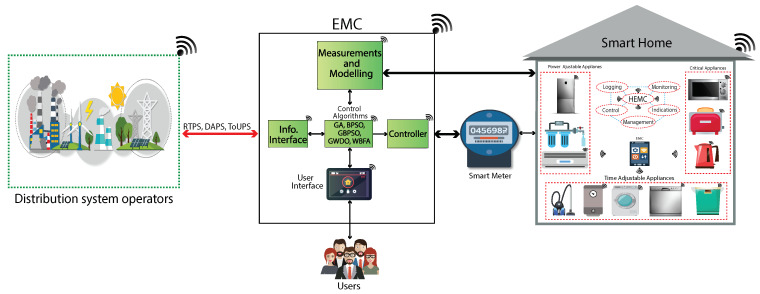
Proposed schematic energy management framework for residential building using DR programs in IoT-enabled environment of the smart grid (SG).

**Figure 2 sensors-20-03155-f002:**
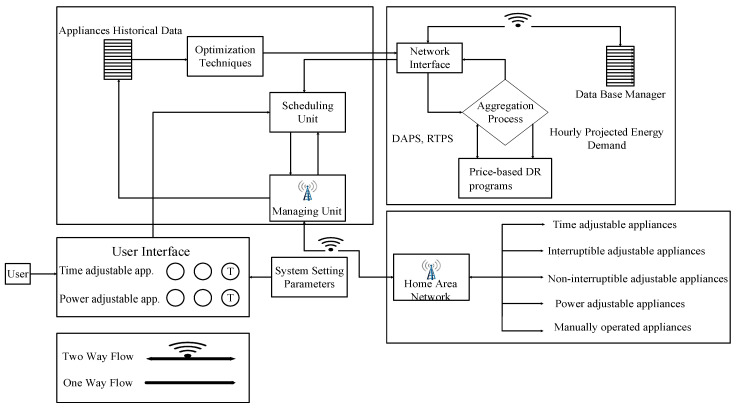
Proposed schematic framework work or functional diagram for energy management via scheduling energy consumption of IoT-enabled smart homes, with the user interface, utility company, and price-based DR modules: The single arrow-head shows one-way flow, and the double arrow-head shows two-way flow.

**Figure 3 sensors-20-03155-f003:**
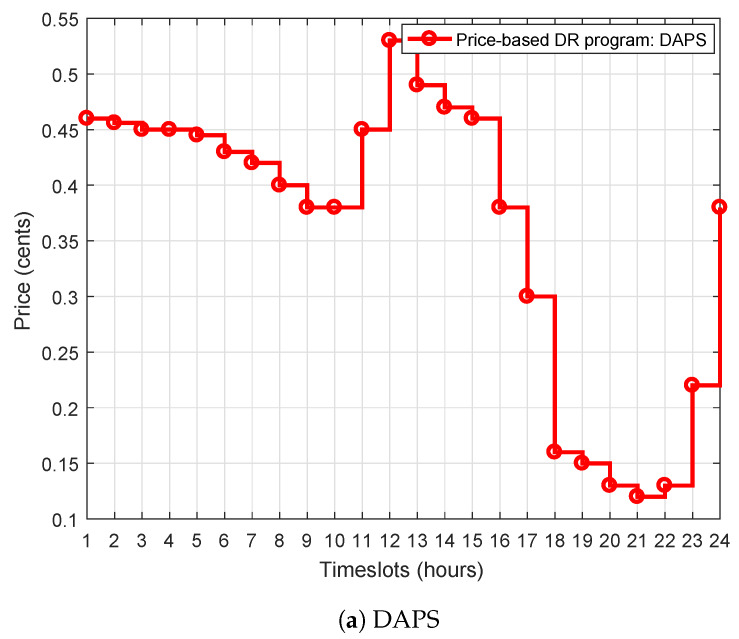
DAPS, RTPS, and ToUPS price-based DR programs adopted from federal energy regulating commission (FERC) for energy management using IoT-enabled environment of the SG.

**Figure 4 sensors-20-03155-f004:**
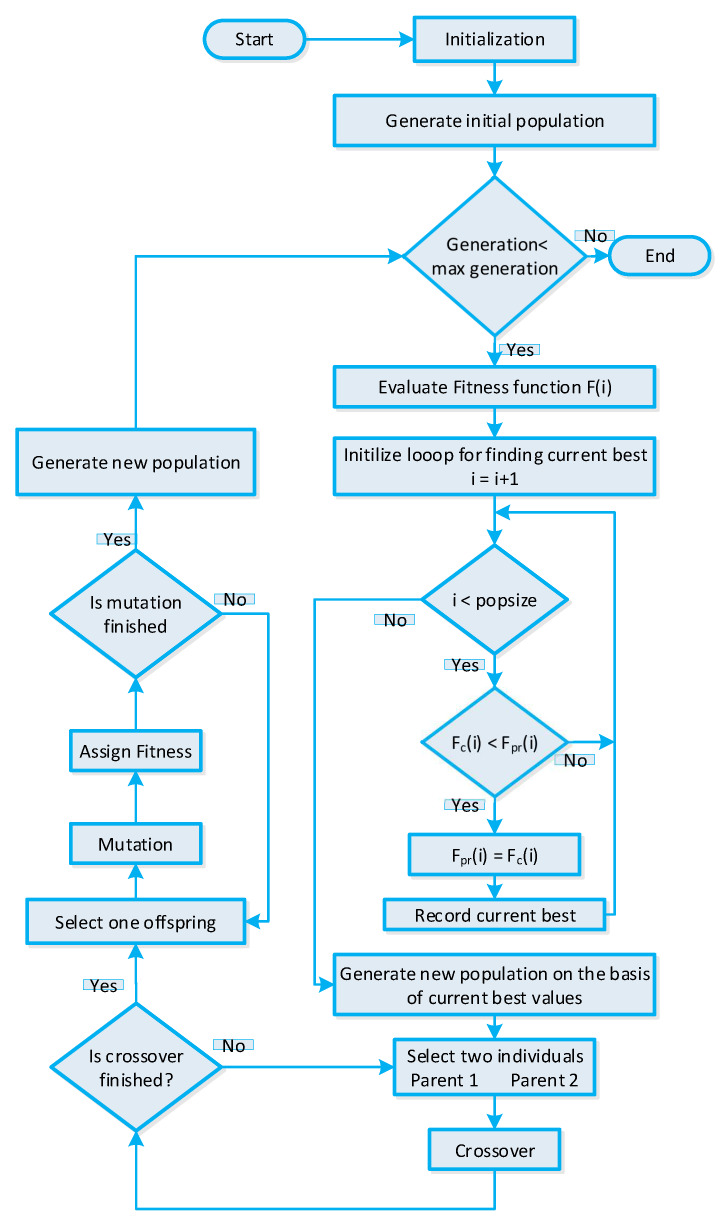
GA-based strategy for efficient energy management of residential building smart appliances in an IoT enabled environment of the SG.

**Figure 5 sensors-20-03155-f005:**
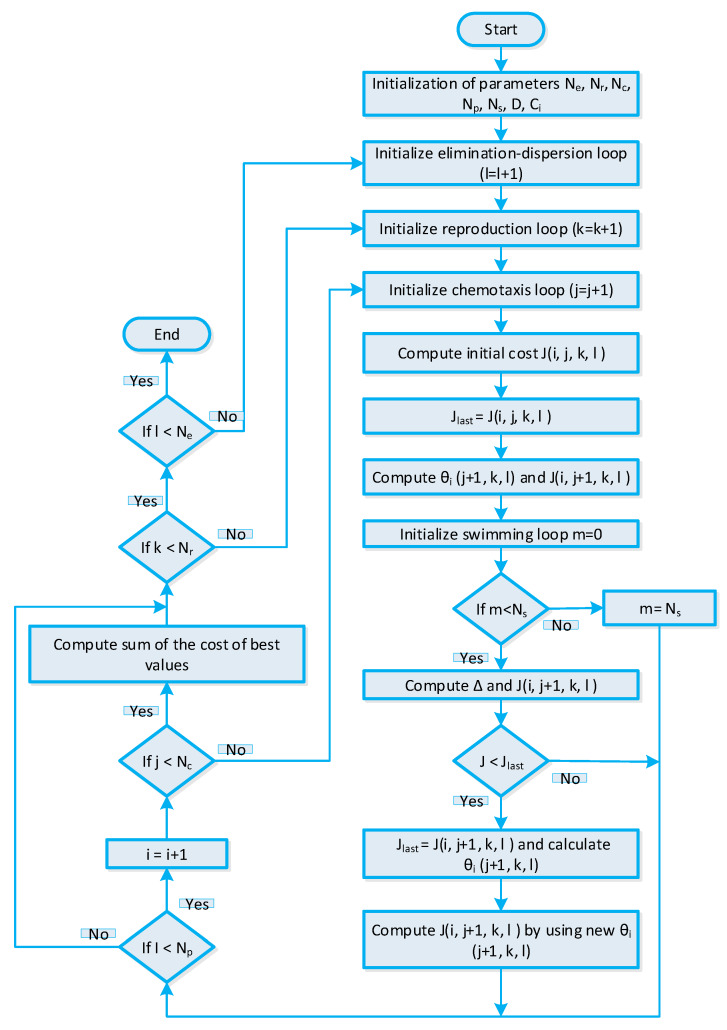
BFO algorithm-based strategy for efficient energy management of residential building smart appliances in an IoT enabled environment of the SG.

**Figure 6 sensors-20-03155-f006:**
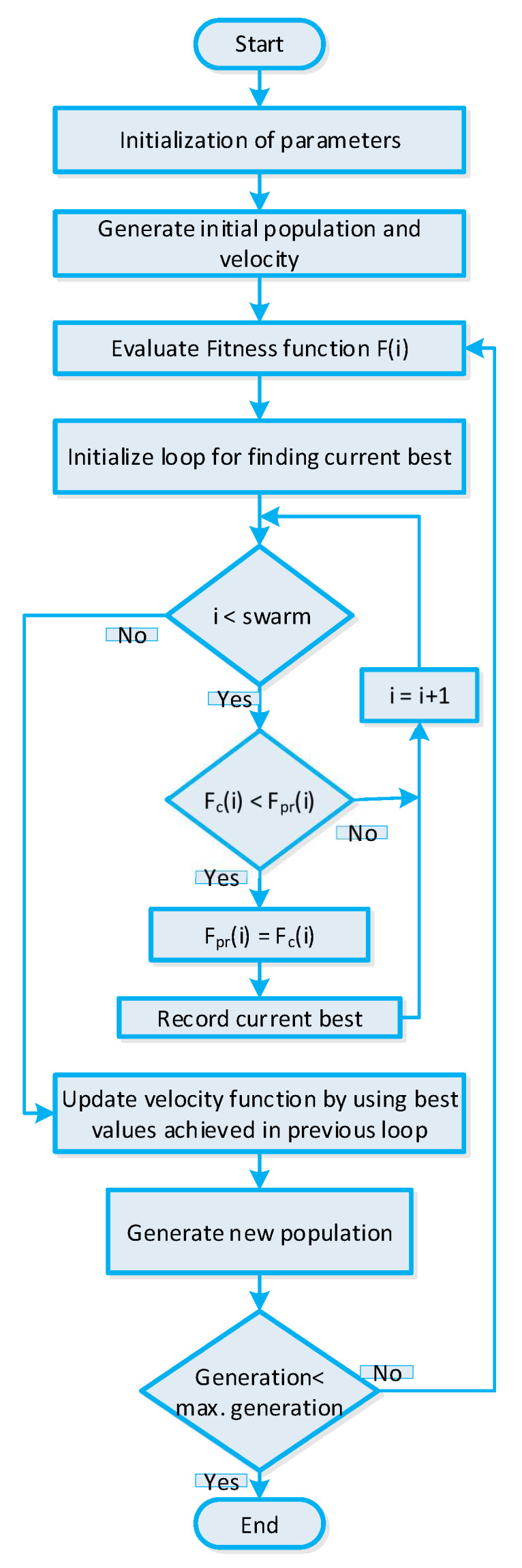
BPSO algorithm-based strategy for efficient energy management of residential building smart appliances in an IoT enabled environment of the SG.

**Figure 7 sensors-20-03155-f007:**
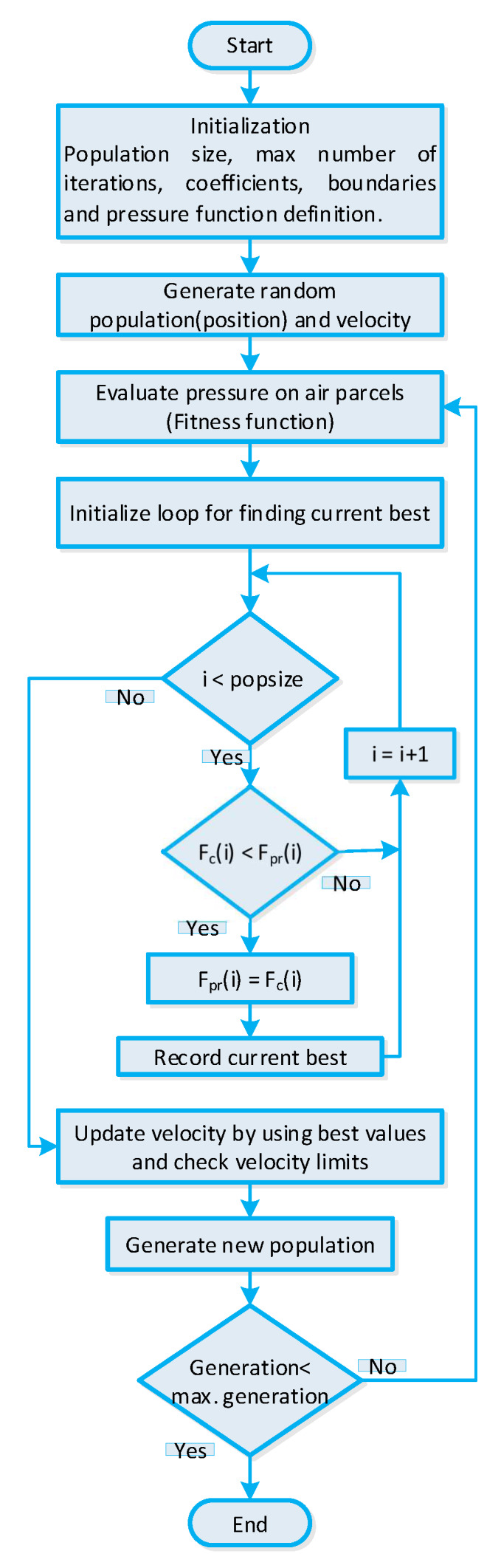
WDO algorithm-based strategy for efficient energy management of residential building smart appliances in an IoT enabled environment of the SG.

**Figure 8 sensors-20-03155-f008:**
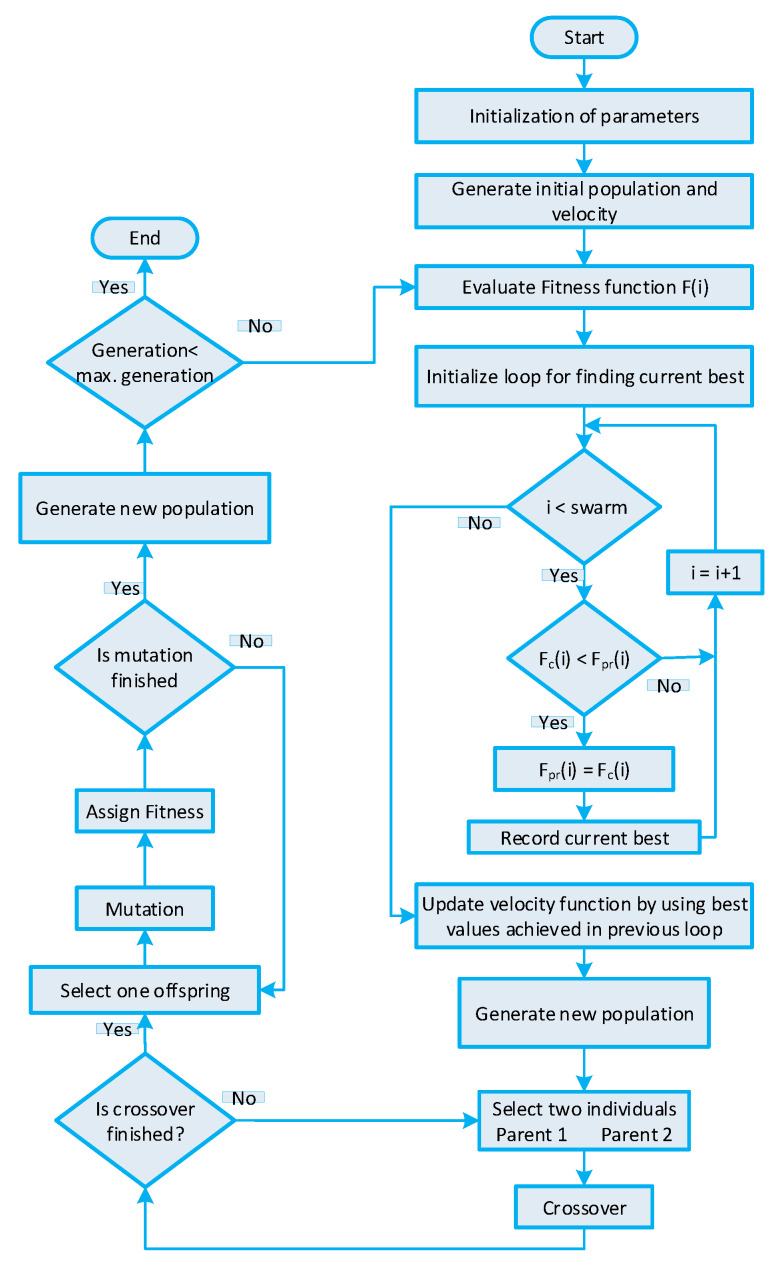
GBPSO algorithm-based strategy for efficient energy management of residential building smart appliances in an IoT enabled environment of the SG.

**Figure 9 sensors-20-03155-f009:**
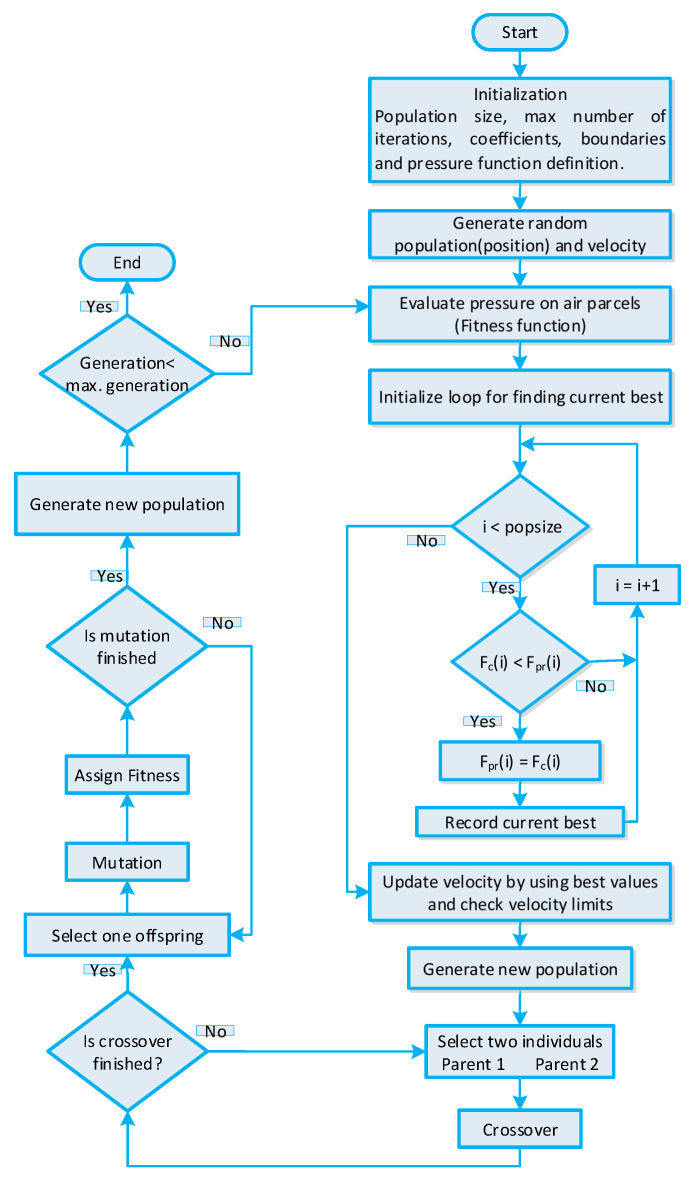
GWDO algorithm-based strategy for efficient energy management of residential building smart appliances in an IoT enabled environment of the SG.

**Figure 10 sensors-20-03155-f010:**
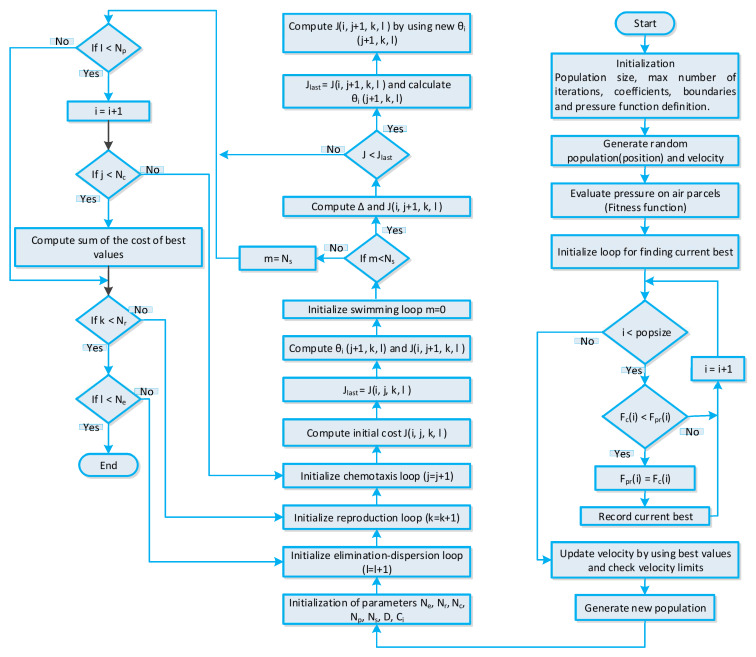
WBFA-based strategy for efficient energy management of residential building smart appliances in an IoT enabled environment of the SG.

**Figure 11 sensors-20-03155-f011:**
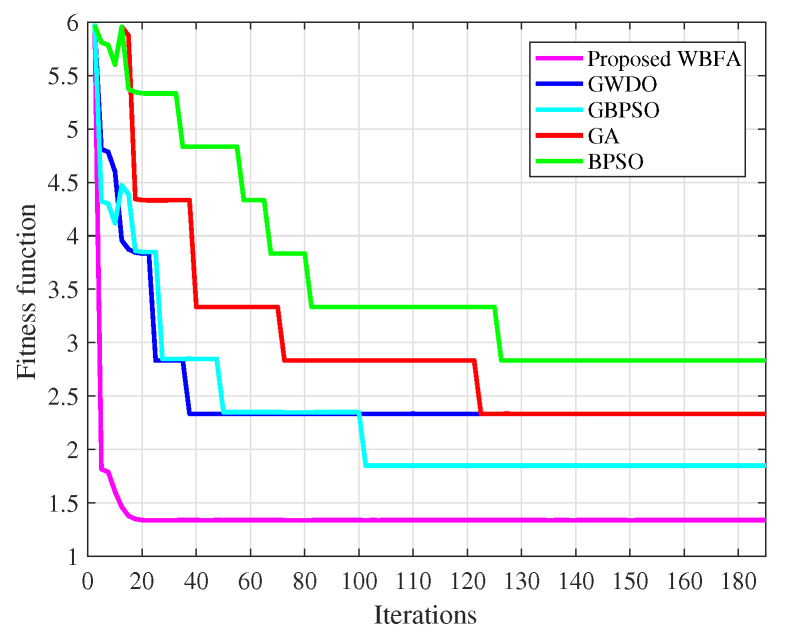
Proposed and adapted algorithms convergence speed analysis.

**Figure 12 sensors-20-03155-f012:**
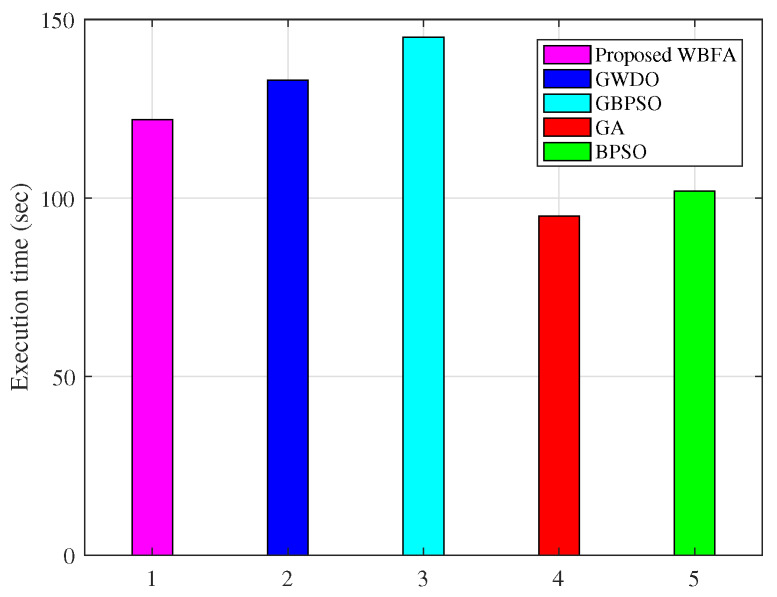
Proposed and adapted algorithm execution times analysis.

**Figure 13 sensors-20-03155-f013:**
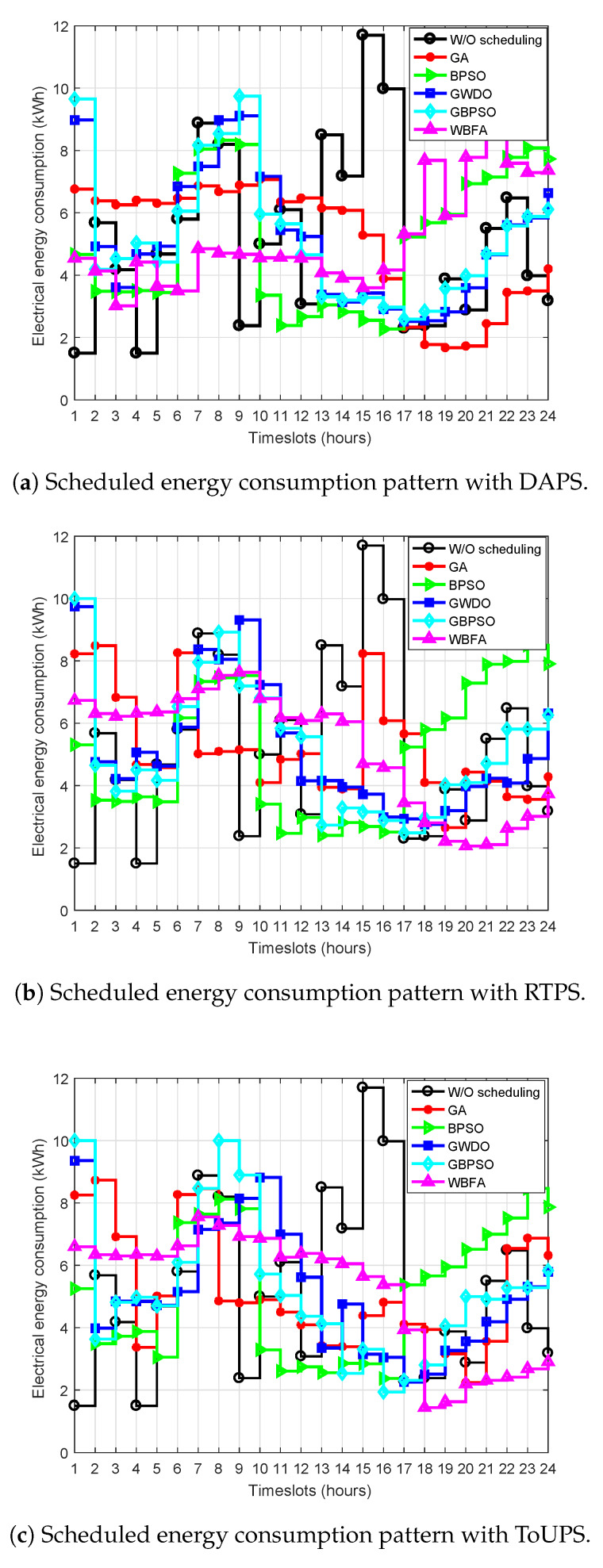
Power usage of residential building smart appliances before and after scheduling using DR programs in IoT-enabled environment of the SG.

**Figure 14 sensors-20-03155-f014:**
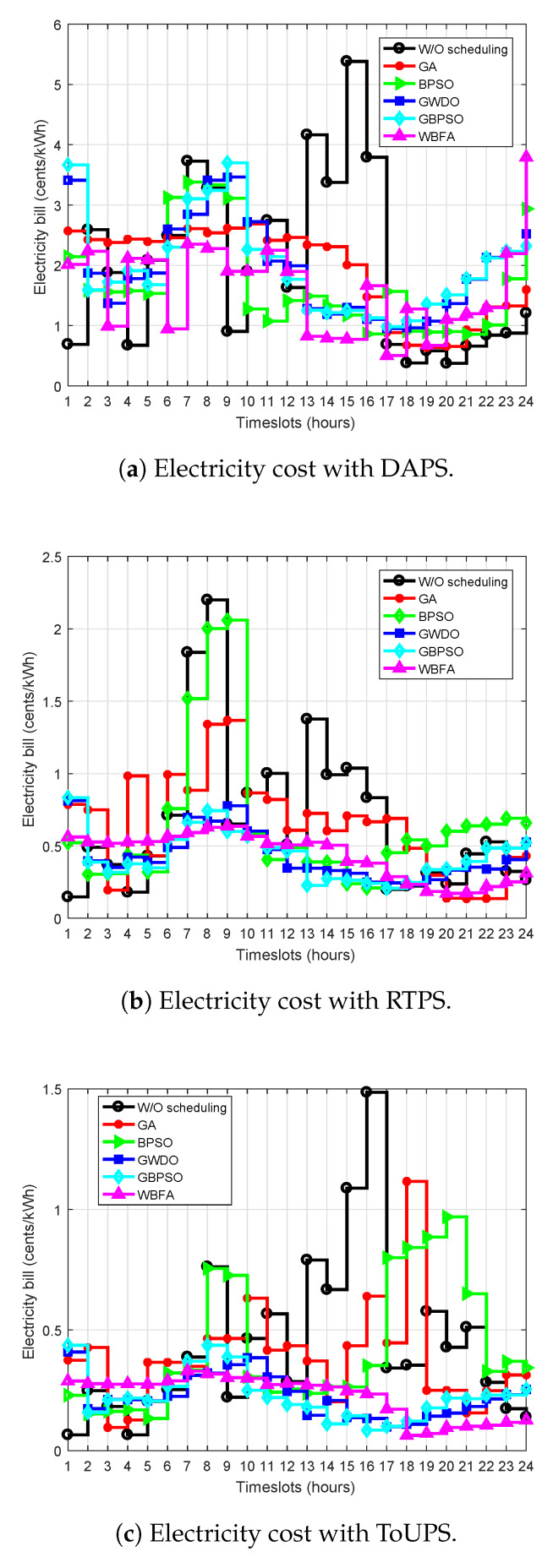
Cost of electricity per hour evaluation before and after scheduling of the proposed WBFA-based strategy and existing strategies using DR programs in IoT-enabled environment of the SG.

**Figure 15 sensors-20-03155-f015:**
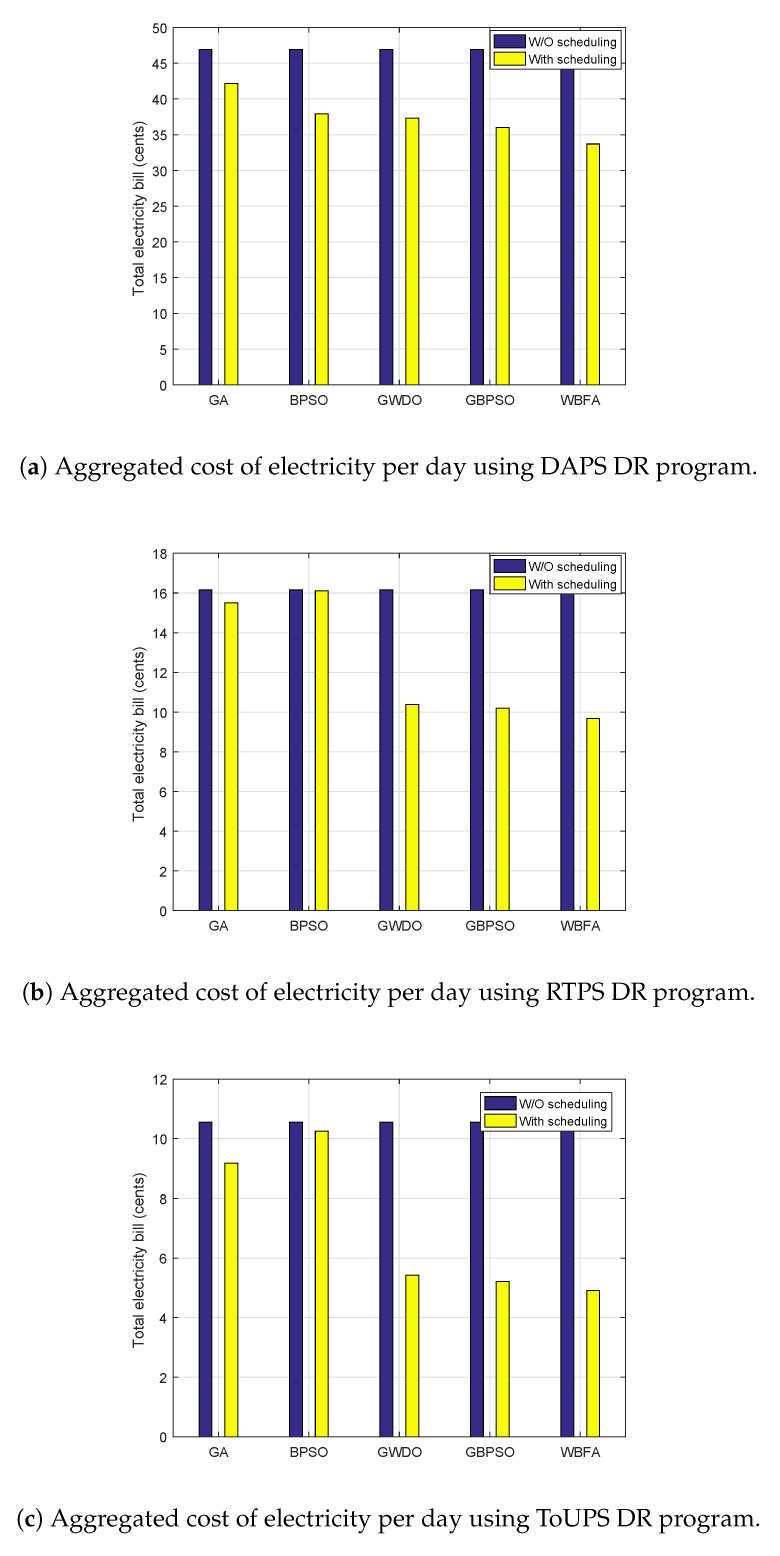
Aggregated cost of electricity per day evaluation before and after scheduling using DR programs in IoT-enabled environment of the SG.

**Figure 16 sensors-20-03155-f016:**
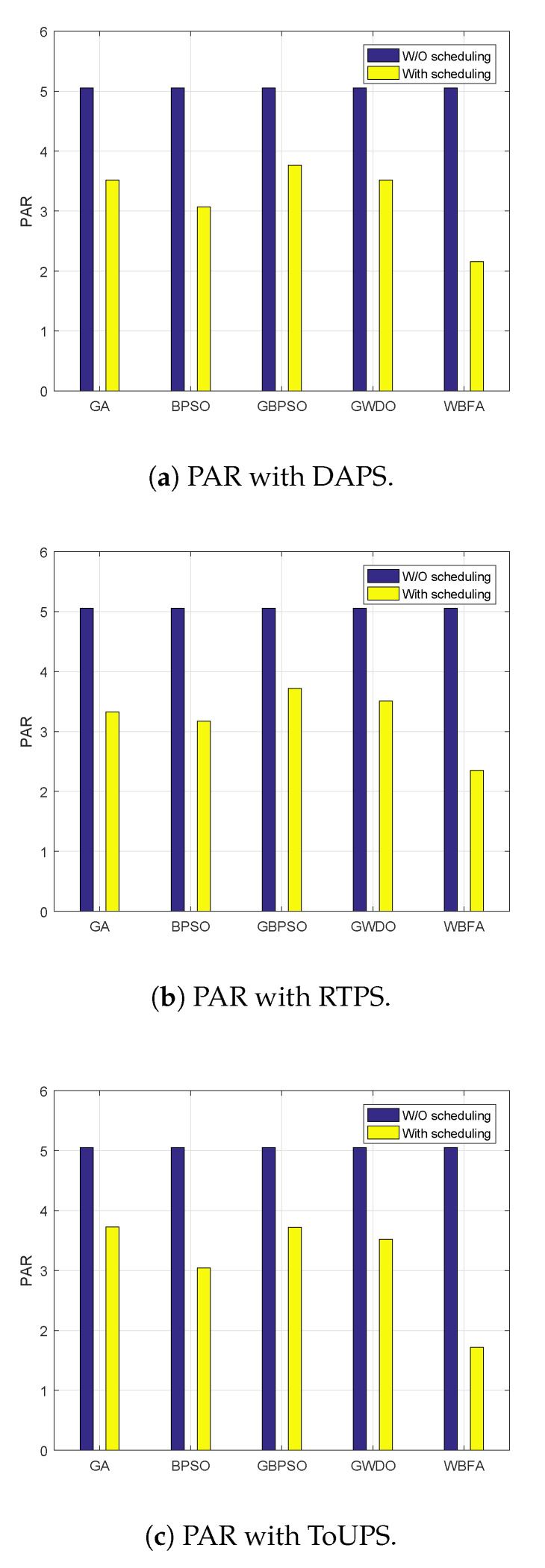
Aggregated PAR per day evaluation before and after scheduling using DR programs in IoT-enabled environment of the SG.

**Figure 17 sensors-20-03155-f017:**
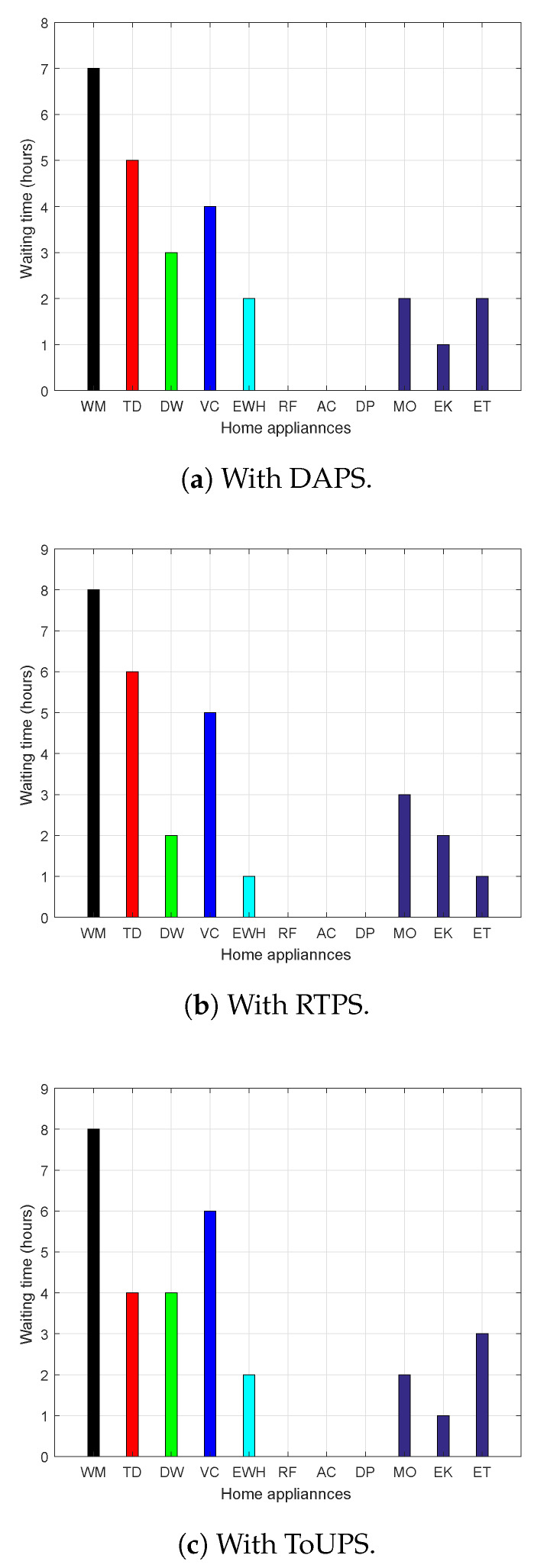
Evaluation of UC in terms of waiting per day after scheduling with proposed WBFA based strategy using price-based DR programs (TD, tumble dryer; WM, washing machine; DW, dishwasher; VC, vacuum cleaner; RF, refrigerator; EWH, electric water heater; AC, air conditioner; DP, dispenser; MO, microwave oven; EK, electric kettle; and ET, electric toaster).

**Figure 18 sensors-20-03155-f018:**
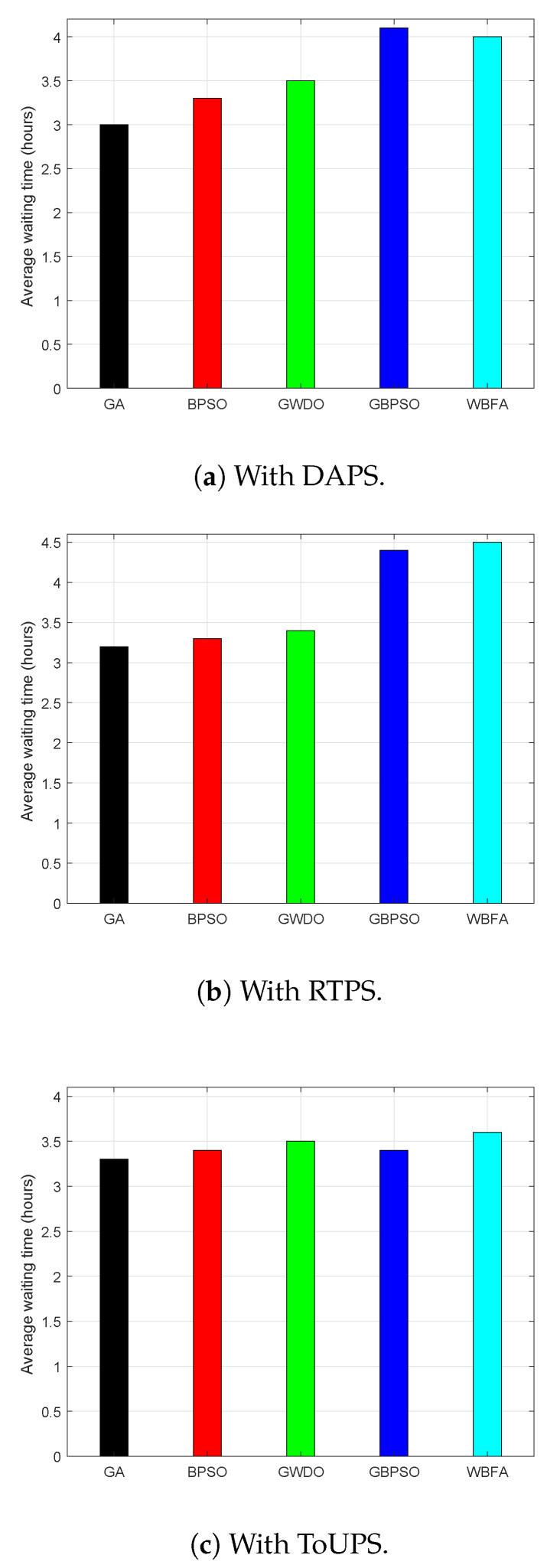
Evaluation of UC in terms of waiting time per day before and after scheduling of the proposed WBFA-based strategy and existing strategies using DR programs in IoT-enabled environment of the SG.

**Table 1 sensors-20-03155-t001:** A summary of mathematical methods-based energy management strategies for residential building in terms of objectives, techniques, demand response (DR) programs, appliances categorization, and limitations.

Energy Management Studies	Techniques	DR Programs	Appliances Categorization	Objectives	Limitations
Appliances and energy storage scheduling for DSM [20]	LP	RTPS	General household appliances	Maximizing consumers savings	UC is compromised while maximizing monetary savings
Optimal household appliance scheduling [21]	MILP	ToUPS	Shiftable load, weather based load, and interruptible load	Minimizing electricity bill	The electricity bill is reduced at the expense of increased system complexity
Residential appliance scheduling via home energy management with DERs and appliance scheduling (HEMDAS) [22]	MINLP	RTPS and fixed price	Controllable thermal appliances, controllable electrical appliances, and appliances	Achieving trade-off between energy cost and inconvenience	PAR is ignored, which is barer for achieving the desired trade-off
Home energy management via home area network (HAN) [23]	MINLP	Dynamic pricing and automatic DR	Deferrable, curtailable, thermal, and critical appliances	Maximizing both savings and UC	Objectives are achieved at the cost of high system complexity
Residential load scheduling [24]	Fractional programming tools	RTPS	Improving cost efficiency	The cost-efficient solution is obtained at the expense of UC	Complexity is increased
Smart heating and air conditioning for Home energy management [25]	ToUPS, RTPS, and critical peak pricing scheme (CPPS)	MINLP and PMIP	Maximizing both consumers convenience and savings	Energy bill reduction	Objectives are achieved at the cost of high system complexity
Automatic scheduling of home appliances [26]	PMIP, MILP, and model predictive control (MPC)	RTPS	Thermal and nonthermal	Minimizing electricity bill and peak power consumption	Objectives are obtained at the cost of UC
Deferrable appliances and energy resources scheduling [27]	MINLP and MPC	RTPS	Deferrable appliances and Non-deferrable appliances	Maintaining the balance between demand and supply	The balance between supply and demand is maintained at increased capital cost
Selfish and cooperative building energy management [28]	EMPC	RTPS	Heat pumps	Minimizing electricity cost	Electricity cost is reduced for both selfish and cooperative buildings while UC is ignored, which is tightly linked with electricity cost
Modular energy management system for urban areas [29]	MILP	Urban area energy systems	Thermal, heating, and cooling prosumers	Maximizing annual cost savings	System complexity is increased while maximizing annual savings

**Table 2 sensors-20-03155-t002:** A summary of heuristic and hybrid methods-based energy management strategies for residential smart homes in terms of objectives, techniques, DR programs, appliances categorization, and limitations.

Energy Management Studies	Techniques	DR Programs	Appliances Categorization	Objectives	Limitation
Optimal energy management in micro-grid [30]	Glowworm swarm particles optimization algorithm	Direct load control (DLC)	Shiftable load	Reducing number of optimization variables and adoption of real valued optimization methods	Increased model complexity
Optimal learning-based energy management system [31]	Heuristic algorithms	DAPS	Regulatable load, fixed load, and deferrable load	Reducing electricity bill payment and peak power consumption	Increased computational overhead
Heuristic optimization towards DSM [32]	EA	DAPS	Regulateable load, fixed load, and deferrable load	Reducing peak load and reshaping load profile	Rebound peaks may be created while achieving the objectives
An optimal household appliances scheduling [38,39]	GA	RTPS-IBRS	Regulateable load, fixed load, and deferrable load	Reducing electricity cost and PAR	Objectives are achieved at the cost of increased system complexity
Realistic residential load scheduling [42,43,45]	GA, GWDO, BPSO, and WDO	RTPS-IBRS	Critical, interruptible, and non-interruptible	Reducing electricity cost and PAR	Objectives are achieved at the cost of increased system complexity
Residential load scheduling towards DSM [48,49,50,51]	GA, teacher learning based optimization (TLBO) algorithm, ILP and HGGS algorithm	RTPS, IBRS, ToUPS, and CPPS	Critical, interruptible, and non-interruptible	Reducing electricity cost, PAR, and discomfort	Objectives are achieved at raising system complexity

**Table 3 sensors-20-03155-t003:** Description of smart appliances parameters equipped with residential building like duration, category, operation time interval, priority, and power rating (WM, washing machine; TD, tumble dryer; DW, dish washer; VC, vacuum cleaner; RF, refrigerator; EWH, electric water heater; AC, air conditioner; WDP, water dispenser; MO, microwave oven; EK, electric kettle; and ET, electric toaster).

Category	Appliances	Duration (Hours/Day)	Power (kW)	Operation Start Time	Operation End Time	Priority
Time adjustable appliances						4
Interruptible time adjustable appliances	WM	6	3.0	9:00 am	3:00 pm	
	TD	4	3.3	4:00 pm	8:00 pm	
	DW	2	2.5	10:00 pm	12:00 am	
Non-interruptible time adjustable appliances						3
	VC	2	1.5	10:00 am	12:00 pm	
	EWH	3	1.8	5:00 am	8:00 am	
Power adjustable appliances						2
	RF	24	0.5–1.5	8:00 am	8:00 am	
	AC	24	0.8–1.8	8:00 am	8:00 am	
	WD	24	0.5–2.0	8:00 am	8:00 am	
Critical appliances						1
	MO	2	1.2	2:00 pm	4:00 pm	
		2	1.9	8:00 am	1:00 am	
	EK	2	1.9	4:00 pm	6:00 pm	
		2	1.9	8:00 pm	10:00 pm	
		2	1.2	7:00 am	9:00 am	
	ET	2	1.2	1:00 pm	3:00 pm	
		2	1.2	8:00 pm	10:00 pm	

**Table 4 sensors-20-03155-t004:** Control parameters and decision variables of the proposed and benchmark strategies for efficient energy management of residential buildings in IoT enabled environment of the SG.

Proposed and Benchmark Algorithms													
**BFO**		**GA**		**BPSO**		**WDO**		**GBPSO**		**GWDO**		**Proposed WBFA**	
						**Parameters**	**Value**						
Pop	100	Pop	100	Pop	100	Pop	100	Pop	100	Pop	100	Pop	100
Itr	200	Itr	200	Itr	200	Itr	200	Itr	200	Itr	200	Itr	200
*n*	11.0	*n*	11.0	*n*	11.0	*n*	11.0	*n*	11.0	*n*	11.0	*n*	11.0
$e_{s}$	24.0	$p_{c}$	0.90	$c_{1}$	2.00	RT	3.00	$c_{1}$	2.00	RT	3.00	RT	3.00
$r_{s}$	5.00	$p_{m}$	0.10	$c_{2}$	2.00	*g*	0.20	$c_{2}$	2.00	*g*	0.20	*g*	0.20
$c_{s}$	5.00	—	—	$w_{i}$	2.00	α	0.40	$w_{i}$	2.00	α	0.40	α	0.40
$s_{s}$	2.00	—	—	$w_{f}$	0.40	$dim_{min}$	−5.00	$w_{f}$	0.40	$dim_{min}$	−5.00	$dim_{min}$	−5.00
$C_{i}$	0.01	—	—	$v_{min}$	−4.00	$dim_{max}$	5.00	$v_{min}$	−4.00	$dim_{max}$	5.00	$dim_{max}$	5.00
$p_{ed}$	0.50	—	—	$v_{max}$	4.00	$v_{min}$	−0.30	$v_{max}$	4.00	$v_{min}$	−0.30	$v_{min}$	−0.30
$C_{i}$	0.98	—	—	—	—	$v_{max}$	0.30	$p_{c}$	0.90	$v_{max}$	0.30	$v_{max}$	0.30
—	—	—	—	—	—	—	—	$p_{m}$	0.10	$p_{c}$	0.90	$e_{s}$	24.0
—	—	—	—	—	—	—	—	—	—	$p_{m}$	0.10	$r_{s}$	5.00
—	—	—	—	—	—	—	—	—	—	—	—	$c_{s}$	5.00
—	—	—	—	—	—	—	—	—	—	—	—	$s_{s}$	2.00

**Table 5 sensors-20-03155-t005:** Evaluation of the proposed WBFA algorithm and adopted algorithms such as GWDO, GBPSO, GA, and BPSO in terms of complexity, convergence rate, and execution time.

Performance Metrics	Proposed and Adopted Algorithms				
	BPSO	GA	GBPSO	GWDO	Proposed WBFA
Complexity (level)	Low	Low	High	High	Moderate
Convergence speed (iterations)	128th	121th	100th	39th	20th
Execution time (s)	102	95	145	133	122

**Table 6 sensors-20-03155-t006:** Aggregated cost of electricity per day evaluation before and after scheduling using DAPS DR program in IoT-enabled environment of the SG.

Parameters	Proposed and Benchmark Strategies with DAPS				
	GA	BPSO	GBPSO	GWDO	WBFA
Electricity cost	42	37	36	35	34
Difference	5	10	11	12	13
Decrement (%)	10	21	23	25	27

**Table 7 sensors-20-03155-t007:** Aggregated cost of electricity per day evaluation before and after scheduling using RTPS price-based DR program in IoT-enabled environment of the SG.

Parameters	Proposed and Benchmark Strategies with RTPS				
	GA	BPSO	GBPSO	GWDO	WBFA
Electricity bill	15	16	10.3	10.1	9.5
Difference	1.1	0.1	5.8	6	6.6
Decrement (%)	6.8	0.62	36	37	40

**Table 8 sensors-20-03155-t008:** Aggregated cost of electricity per day evaluation before and after scheduling using ToUPS price-based DR program in IoT-enabled environment of the SG.

Parameters	Proposed and Benchmark Strategies with ToUPS				
	GA	BPSO	GBPSO	GWDO	WBFA
Electricity cost	9.2	10.1	5.4	5.2	4.9
Difference	1.35	0.45	5.15	5.35	5.56
Decrement (%)	12.7	4.2	48.1	50.2	52.1

**Table 9 sensors-20-03155-t009:** Evaluation of aggregated PAR per day before and after scheduling using DAPS price-based DR program.

Parameters	Proposed and Benchmark Strategies with DAPS				
	GA	BPSO	GBPSO	GWDO	WBFA
PAR	3.4	3.1	3.8	3.6	2.1
Difference	1.6	1.9	1.2	1.4	2.9
Decrement (%)	32.0	38.0	24.0	28.0	58.0

**Table 10 sensors-20-03155-t010:** Evaluation of aggregated PAR per day before and after scheduling using RTPS price-based DR program.

Parameters	Proposed and Benchmark Strategies with RTPS				
	GA	BPSO	GBPSO	GWDO	WBFA
PAR	3.3	3.1	3.6	3.5	2.3
Difference	1.7	1.9	1.4	1.5	2.7
Decrement (%)	34.0	38.0	28.0	20.0	54.0

**Table 11 sensors-20-03155-t011:** Evaluation of aggregated PAR per day before and after scheduling using ToUPS price-based DR program.

Parameters	Proposed and Benchmark Strategies with ToUPS				
	GA	BPSO	GBPSO	GWDO	WBFA
PAR	3.8	3.0	3.9	3.70	1.9
Difference	1.2	2.0	1.1	1.3	3.1
Decrement (%)	24.0	40.0	22.0	26.0	62.0

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
