# Peer review of "Efficient Energy Management of IoT-Enabled Smart Homes Under Price-Based Demand Response Program in Smart Grid"

_sensors, 2020, doi:10.3390/s20113155_

Round 1
Reviewer 1 Report
I recommend the paper to be accepted with a minor revision that addresses the below comments.
1) The authors have presented the rationale for selecting WBFA in Section 4.1. Some of these rationales include lower computational time and cost. It would be complementary to their study if they include a table or a graph that shows quantitatively how WBFA satisfies the rationale. For example, can WBFA's lower computational time be compared with other methods' in the form of convergence time or utilization of memory or resources?
2) It is evident through the results that WBFA is performing significantly better. However, it is unclear why it is doing better. Is it because WBFA captures the problem better than other methods in the form of parameters as shown in Table 4? It would be good if the authors can add this to their discussion in Section 6
3) In Section 7, a commentary on the application scenarios for EMCs would be useful. What modifications would the WBFA-based EMC need (in terms of additional parameters, etc.) when renewables and microgrids are considered? Although the authors mention this as the first point in the future work, an additional explanation can be included since it is one of the fastest emerging use-cases in DR and transactive energy markets
Author Response
I recommend the paper to be accepted with a minor revision that addresses the below comments.
Comment 1: The authors have presented the rationale for selecting WBFA in Section 4.1. Some of these rationales include lower computational time and cost. It would be complementary to their study if they include a table or a graph that shows quantitatively how WBFA satisfies the rationale. For example, can WBFA's lower computational time be compared with other methods' in the form of convergence time or utilization of memory or resources?
Author Response: The authors are highly grateful to the respected Reviewer for his suggestions regarding adapted and proposed algorithms. As per your valuable suggestion, the table and graph both are included that reflect adapted and proposed algorithms behavior in terms of convergence speed, execution time, and complexity. Please track changes in section 5.
Author Action: The changes have been made in Section 5 of the revised manuscript. Please check Figures 10, 11, and Table 5.
Comment 2: It is evident through the results that WBFA is performing significantly better. However, it is unclear why it is doing better. Is it because WBFA captures the problem better than other methods in the form of parameters as shown in Table 4? It would be good if the authors can add this to their discussion in Section 6.
Author Response: Respected Reviewer, thank you very much for your suggestion. As per your suggestion, the solid reasoning of why WBFA outperforms existing methods is incorporated in Section 5. Please have a look at Section 5.
Author Action: The suggested changes are made in the reasoning of the proposed WBFA algorithm. Please check Section 5.
Comment 3: In Section 7, a commentary on the application scenarios for EMCs would be useful. What modifications would the WBFA-based EMC need (in terms of additional parameters, etc.) when renewables and microgrids are considered? Although the authors mention this as the first point in the future work, an additional explanation can be included since it is one of the fastest emerging use-cases in DR and transactive energy markets
Author Response: Dear Reviewer, thank you very much for such a valuable suggestion. As per your valuable suggestion, the commentary on applications and modifications of WBFA-based EMC is included for considering renewables, microgrids, EVs, energy storage systems, and hybrid systems. Also, it is highlighted as an emerging technology to participate in the regulated energy market to facilitate both the power grid and consumers. Please check Section 7.
Author Action: The suggested changes are made in the first point of Section 7. Please track changes.

Reviewer 2 Report
This study introduces an efficient energy management scheme for smart homes under price-based demand response program. A hybrid of wind-driven and bacterial foraging optimization algorithm is proposed to reduce the peak load and minimize the energy cost.
Detailed comments are as follows:
1. The energy management at home is conducted by an embedded system with limited computational power. As a result, the computational complexity of the proposed scheme has to be analyzed, such as the time complexity, space complexity, CPU time, etc.
2. What are the computational platform is used in this research? And what are the scales of the optimization algorithm (like the number of decision variables, the number of constraints, etc.)? These kinds of information must be provided for an optimization research.
3. Sections 5.1 to 5.6 are not necessary to be included in the manuscript. These existing algorithms have been addressed in Section 2 - related work. The readers can referr to given references for detailed information.
Author Response
This study introduces an efficient energy management scheme for smart homes under price-based demand response program. A hybrid of wind-driven and bacterial foraging optimization algorithm is proposed to reduce the peak load and minimize the energy cost.
Detailed comments are as follows:
Comment 1: The energy management at home is conducted by an embedded system with limited computational power. As a result, the computational complexity of the proposed scheme has to be analyzed, such as the time complexity, space complexity, CPU time, etc.
Author Response: Respected Reviewer, thank you very much for your critical comment. The proposed and adapted energy management strategies are analyzed in terms of computational complexity, convergence speed, and CPU time in Table 5. Please check Section 5.
Author Action: The suggested changes are in the revised manuscript. Please track changes in Section 5.
Comment 2: What are the computational platform is used in this research? And what are the scales of the optimization algorithm (like the number of decision variables, the number of constraints, etc.)? These kinds of information must be provided for an optimization research.
Author Response: Respected Reviewer, thank you very much for your critical comment. The response of your valuable comment is incorporated in Sections 4, 5, and 6, respectively. The control parameters and decision variables are listed in Table 4, 5, and constraints for the optimization problem are referred to in Equations 22a, 22b, 22c, and 22d. Please check
Author Action: The manuscript is revised according to your comments. Please track changes in the revised manuscript.
Comment 3: Sections 5.1 to 5.6 are not necessary to be included in the manuscript. These existing algorithms have been addressed in Section 2 - related work. The readers can referr to given references for detailed information.
Author Response: Respected Reviewer, thank you very much for your valuable suggestions. Yes, I agree with this comment and know the existing strategies are referred to in the related work Section 2. However, the existing strategies are discussed in Section 5 are those with which our proposed strategy is compared for validation purpose.
Author Action: The suggested changes are made in the revised manuscript. Please track changes in the revised manuscript.

Reviewer 3 Report
The paper studied the problem of scheduling energy consumption pattern of smart home appliances in order to mitigate electricity bill, alleviate PAR, and minimize user discomfort.
The topic is very interesting and timely. The paper is well-written and easy to follow which is the most important aspect of the paper. There exist minor typos in the paper which should be modified in case of acceptance.
Parameters in Eqs. (3) to (8), (17) and (21) are not defined! It makes the evaluation of the formulation difficult.
Among the three categories of the appliances presented in this work, only one of them contains the flexible power rating. What happens if an appliance is adjustable power and time simultaneously? How you deal with this kind of appliances?
In lines 254 and 255, the order of gamma parameters are not correct. The one with the highest value should represent on-peak period.
The description in lines 278 and 279, "The strategy adopted has the problem of unguided mutation due to which load becomes unbalanced." is unclear.
Using Eq. (15), how you exactly model flexible power rating? In other words, what is the difference between Eq. (15) and Eqs. (10) and (12)?
Line after Eq. (17) refers to a wrong cross-reference.
In line 300, the word rates is missing after "represent RTPS and ToUPS".
One of the weak aspect of this work is the "User comfort and discomfort". Considering "the user comfort in terms of waiting" is an extensively simplified model. As the authors mentioned in lines 309 and 310, the user comfort is a factor of several parameters. It is not only depends on energy consumption, waiting time, temperature, illumination, air quality, humidity, and sound, but also depends on the demographic profile of the residents. There exist several paper which discuss the user behavior modeling in smart environments, for example see (Khamesi, Atieh R., et al. "Perceived-Value-driven Optimization of Energy Consumption in Smart Homes." ACM Transactions on Internet of Things 1.2 (2020): 1-26.) Again, parameters in Eq. (21) are not defined so the reviewer is not able to evaluate it.
The novelty of the proposed solution is questionable. As mentioned in lines 426 and 427 "The WBFA is a hybrid algorithm of WDO and BFA algorithms." Given the simulation results, the reviewer is not convinced that the proposed solution makes a significant improvement.
The layout of the Table 4 is problematic.
In lines 464 and 465, the authors claimed that "it is clearly noticed that the scheduled energy consumption profile of our proposed scheme is optimal." which is a big assertion and there is no support for it.
Given the results depicted in Fig. 16, the proposed method clearly sacrifices the user comfort, however, in line 469, it is concluded that "electricity is reduced without compromising their comfort." This is inconsistent.
In Fig. 10, What is the difference between 10a and 10b? Both DAPS, RTPS give the same results! Besides, the sub-figure captions are cut which seems there is an overlap between the figures.
Figs. 11 and 12 are very unorganized. It is not clear why the authors believed that their proposed solution outperforms others!
Line 503, it should be Figure 13b.
Line 521, the cross-reference is missing.
In lines 554 and 555 of the conclusion, it is mentioned that "We introduced three price-based DR programs like DAPS, RTPS, and ToUPS." However, they are adopted from other works.
Again, in line 558, the authors assert that "WBFA is to reduce electricity bill, PAR, and without compromising user comfort." which is not accurate.
Author Response
The paper studied the problem of scheduling energy consumption pattern of smart home appliances in order to mitigate electricity bill, alleviate PAR, and minimize user discomfort.
The topic is very interesting and timely. The paper is well-written and easy to follow which is the most important aspect of the paper. There exist minor typos in the paper which should be modified in case of acceptance.
Comment 1: Parameters in Eqs. (3) to (8), (17) and (21) are not defined! It makes the evaluation of the formulation difficult.
Author Response: The authors are highly grateful to the respected Reviewer for his suggestions regarding define parameters of Equation. As per your valuable suggestion, the parameters of Eqs. (3) to (8), (17) and (21) are defined. Please have a look at the Subsection 3.1.1.
Author Action: The parameters of Eqs. (3) to (8), (17) and (21) are defined in Section 3. Please track changes in the revised manuscript.
Comment 2: Among the three categories of the appliances presented in this work, only one of them contains the flexible power rating. What happens if an appliance is adjustable power and time simultaneously? How you deal with this kind of appliances?
Author Response: Thank you very much for giving us a future research direction. As per your recommendation, in the future we place the 4th category of appliances that have both time and power-adjustable behavior, this category will greatly contribute to cost reduction and have little impact on user-discomfort. Please have a look at the future research direction.
Author Action: This comment is added in future research direction Section 7. Please track changes in Future work.
Comment 3: In lines 254 and 255, the order of gamma parameters is not correct. The one with the highest value should represent on-peak period.
Author Response: Thank you very much for identifying the typos issue. As per your recommendation, the order of gamma parameters is now corrected. Please have look at Section 3.1.
Author Action: The changes have been made in the revised manuscript. Please check the changes in Section 3.1.
Comment 4: The description in lines 278 and 279, "The strategy adopted has the problem of unguided mutation due to which load becomes unbalanced." is unclear.
Author Response: The authors are highly grateful to the respected Reviewer regarding ambiguity in a sentence. The unguided mutation refers to the mutation that applies to a random population rather on the local best population. The mutation that is applied to the randomly generated population has no credibility to return the optimal solution and in some cases, the load also becomes unstable due to randomness in population. For an in-depth understanding, interested candidates are referred to [1].
Author Action: The changes have been made in the revised manuscript. Please check changes in Section 4.
Comment 5: Using Eq. (15), how you exactly model flexible power rating? In other words, what is the difference between Eq. (15) and Eqs. (10) and (12)?
Author Response: Dear Reviewer, thank you very much for your critical comment. The flexibility exists either in terms of time and in power rating and is modeled in Equations 4, 6, 8, and 15, in Sections 3 and 4, respectively. Please have a look at Section 3 and Section 4.
Author Action: The changes have been made in Sections 3 and 4 in the revised manuscript. Please track changes in Sections 3 and 4.
Comment 6: Line after Eq. (17) refers to a wrong cross-reference.
Author Response: Thank you very much for identifying our mistake. The wrong cross-referencing issue is resolved in the revised manuscript. Please check.
Author Action: The wrong cross-referencing issue is resolved in the revised manuscript. Please check Section 4.
Comment 7: In line 300, the word rates is missing after "represent RTPS and ToUPS".
Author Response: Respected Reviewer, thank you very much for identifying typos. The missing word is inserted and the whole manuscript is checked, and all typo issues are resolved.
Author Action: The typo issues in the revised manuscript are resolved. Please check the changes.
Comment 8: One of the weak aspect of this work is the "User comfort and discomfort". Considering "the user comfort in terms of waiting" is an extensively simplified model. As the authors mentioned in lines 309 and 310, the user comfort is a factor of several parameters. It is not only depends on energy consumption, waiting time, temperature, illumination, air quality, humidity, and sound, but also depends on the demographic profile of the residents. There exist several paper which discuss the user behavior modeling in smart environments, for example see (Khamesi, Atieh R., et al. "Perceived-Value-driven Optimization of Energy Consumption in Smart Homes." ACM Transactions on Internet of Things 1.2 (2020): 1-26.) Again, parameters in Eq. (21) are not defined so the reviewer is not able to evaluate it.
Author Response: Respected reviewer, we pay a bundle of thanks for your concern. We agree with your concern that user comfort in terms of waiting time is an extensively simplified model. However, our focus in this work is to perform energy management via household load scheduling. Therefore, the user feels comfort or discomfort when the EMC schedule their predefined energy usage pattern. This comfort or discomfort the users face is in terms of waiting time. That is why we modeled user-comfort in terms of waiting time or delay the user faced after following the schedule provided by EMC. We are keeping in view your precise comment and in the future, we will model user comfort and discomfort in terms of some other parameters discussed in this work. Moreover, the parameters of Equation 21 are now defined in the revised manuscript. Please track changes.
Author Action: The suggested changes are incorporated in the revised manuscript. Please check the changes.
Comment 9: The novelty of the proposed solution is questionable. As mentioned in lines 426 and 427 "The WBFA is a hybrid algorithm of WDO and BFA algorithms." Given the simulation results, the reviewer is not convinced that the proposed solution makes a significant improvement.
Author Response: Respected reviewer, we pay a bundle of thanks for the comment. From the simulation results Section 6, it is obvious that our proposed WBFA algorithm outperforms the existing algorithms in terms of electricity cost reduction, and PAR alleviation. Furthermore, our proposed algorithm reduced the cost by 52.1% and PAR by 62.2%, this reduction is significant as compared to the existing methods. The detailed evaluation of our proposed algorithm in terms of achievements is listed in simulation results Section 6. Please have a look at the simulation section.
Author Action: The achievements of our proposed algorithm are listed in Section 5 and Section 6. Please check.
Comment 10: The layout of the Table 4 is problematic.
Author Response: Respected Reviewer, thank you very much for identifying formatting issues. The issue related to the layout of Table 4 is resolved. Please track the changes.
Author Action: The Table 4 formatting issue is resolved in the revised manuscript. Please check Table 4.
Comment 11: In lines 464 and 465, the authors claimed that "it is clearly noticed that the scheduled energy consumption profile of our proposed scheme is optimal." which is a big assertion and there is no support for it.
Author Response: Respected Reviewer, thank you very much for your critical comment. Figures 13 and 14 are in support of this assertion. As in this study, our focus is to shift the load from on-peak hours to off-peak hours by load scheduling, if we look at Figures 13 and 14 keeping in view this statement so our proposed algorithm curve the load is shifted from on-peak hours to off-peak hours. Thus, from the discussion, we came across to the conclusion that our proposed algorithm-based energy consumption curve is optimal because the load is shifted from on-peak hours to off-peak hours.
Author Action: The changes are made in the revised manuscript. Please check the changes.
Comment 12: Given the results depicted in Fig. 16, the proposed method clearly sacrifices the user comfort, however, in line 469, it is concluded that "electricity is reduced without compromising their comfort." This is inconsistent.
Author Response: Respected Reviewer, thank you very much for identifying typos. The sentence is corrected in the revised manuscript. Please have a look simulation result in Section 6.
Author Action: The typos issue is resolved in the revised manuscript. Please check the corrections.
Comment 13: In Fig. 10, What is the difference between 10a and 10b? Both DAPS, RTPS give the same results! Besides, the sub-figure captions are cut which seems there is an overlap between the figures.
Author Response: Respected Reviewer, thank you very much for identifying formatting issues and typos. Both RTPS and DAPS are different, correct figures are placed in the revised manuscript. Now Figure 10 is Figure 12 in the revised manuscript. Issues of figure and sub-caption overlapping are resolved in the revised manuscript. Please have a look at Figure 12.
Author Action: The issues in Figure 12 are resolved in the revised manuscript. Please check the changes in Figure 12.
Comment 14: Figs. 11 and 12 are very unorganized. It is not clear why the authors believed that their proposed solution outperforms others!
Author Response: Respected Reviewer, thank you very much for your critical comment. Figure 11 and Figure 12 now in the revised manuscript are Figures 13 and Figure 14, respectively. These Figures are well organized if you look at the Price-based DR programs Figure 12, whenever there are on-peak hours (high price hours) in Figure 12, our proposed algorithm shifted the load from that on-peak hours to off-peak hours to reduce both electricity cost and PAR.
Author Action: The stable and organized behavior of energy consumption for our proposed algorithm is depicted in Figures 13 and 14, respectively. Please check the changes.
Comment 15: Line 503, it should be Figure 13b.
Author Response: Respected Reviewer, thank you very much for identifying typos. The Figure is now correctly cross-referred and Figure 13b is now Figure 15b in the revised manuscript and. Please have a look at the simulation Section 6.
Author Action: The figure cross-referencing issue is resolved in the revised manuscript. Please check the changes.
Comment 16: Line 521, the cross-reference is missing.
Author Response: Respected Reviewer, thank you very much for identifying typos. The missing cross-referencing issue is resolved in the revised manuscript. Please have a look at the revised manuscript.
Author Action: The typo issue is resolved please check the revised manuscript.
Comment 17: In lines 554 and 555 of the conclusion, it is mentioned that "We introduced three price-based DR programs like DAPS, RTPS, and ToUPS." However, they are adopted from other works.
Author Response: Thank you very much for identifying our mistakes. The correct phrase adopted is in place of introduced in the revised manuscript.
Author Action: The phrase correction issue is resolved. Please have look at the revised manuscript.
Comment 18: Again, in line 558, the authors assert that "WBFA is to reduce electricity bill, PAR, and without compromising user comfort." which is not accurate.
Author Response: Respected reviewer, thank you very much for identifying our mistakes. The correct phrase without compromising user comfort is replaced with sacrificing user comfort. The sentence is corrected in the revised manuscript.
Author Action: Please have a look at the revised manuscript.
Round 2
Reviewer 3 Report
Comments 3 and 8 are not addressed properly. Please elaborate.
Author Response
Reviewer Comments
Paper title: Efficient Energy Management of IoT Enabled Smart Homes Under Price-based Demand Response Program in Smart Grid (ID: sensors-781799)
Date: May 20, 2020
Dear Reviewer, thank you very much for your time and effort to deeply review the paper. Your comments really guided us to improve the quality of the paper up to the level of the journal. Please note that responses to the comments of Reviewer 3 are given in “red” color,that show refinements (grammar, acronyms, etc.) in the revised version of the manuscript.
Regards
Imran Ullah Khan (PhD)
Comment 3: In lines 254 and 255, the order of gamma parameters is not correct. The one with the highest value should represent on-peak period.
Author Response: Thank you very much for your precious comment. As per your suggestion, the order of gamma parameters is now corrected for ToUPS. Furthermore, the ToUPS is elaborated more in the revised manuscript for clear understanding. Please have look at Section 3 and Equation
Author Action: The changes have been made in the revised manuscript. Please check the changes in Section 3.1.
Comment 8: One of the weak aspect of this work is the "User comfort and discomfort". Considering "the user comfort in terms of waiting" is an extensively simplified model. As the authors mentioned in lines 309 and 310, the user comfort is a factor of several parameters. It is not only depends on energy consumption, waiting time, temperature, illumination, air quality, humidity, and sound, but also depends on the demographic profile of the residents. There exist several paper which discuss the user behavior modeling in smart environments, for example see (Khamesi, Atieh R., et al. "Perceived-Value-driven Optimization of Energy Consumption in Smart Homes." ACM Transactions on Internet of Things 1.2 (2020): 1-26.) Again, parameters in Eq. (21) are not defined so the reviewer is not able to evaluate it.
Author Response: Respected reviewer, we pay a bundle of thanks for your concern. We agree with your concern that user comfort in terms of waiting time is an extensively simplified model. However, our focus in this work is to perform energy management via household load scheduling. Therefore, the user feels frustration in terms of delay (waiting time) when the EMC shifts their load from on-peak hours to off-peak hours by rescheduling their predefined operating pattern in response to the price-based DR program. The frustration in terms of delay (waiting time) is termed as user comfort or discomfort. The more delay (waiting time) the users face due load shifting is termed as user discomfort and the less delay the user's face is comfort. The user comfort or discomfort in terms of waiting time is now defined in Equations 21, 22, and 23. We modeled user comfort in terms of waiting time or delay because in scheduling operation of appliances may be delayed or advanced by the EMC and users feel frustration when waits for an activity. We are keeping in view your precise comment and in the future, we will model user comfort and discomfort in terms of some other parameters discussed in this work. Moreover, the parameters of Equation 21 are now defined in the revised manuscript. Please track changes.
Author Action: The suggested changes are incorporated in the revised manuscript. Please check the changes.
